# Independent representations of ipsilateral and contralateral limbs in primary motor cortex

Ethan A Heming[1†], Kevin P Cross[1†*], Tomohiko Takei[1,2], Douglas J Cook[1,3,4], Stephen H Scott[1,5,6]

[1]Centre for Neuroscience Studies, Queen's University, Kingston, Canada; [2]Graduate School of Medicine, The Hakubi Center for Advanced Research, Kyoto University, Kyoto, Japan; [3]Department of Surgery, Queen's University, Kingston, Canada; [4]Department of Surgery, Dalhousie University, Halifax, Canada; [5]Department of Medicine, Queen's University, Kingston, Canada; [6]Department of Biomedical and Molecular Sciences, Queen's University, Kingston, Canada

**Abstract** Several lines of research demonstrate that primary motor cortex (M1) is principally involved in controlling the contralateral side of the body. However, M1 activity has been correlated with both contralateral and ipsilateral limb movements. Why does ipsilaterally-related activity not cause contralateral motor output? To address this question, we trained monkeys to counter mechanical loads applied to their right and left limbs. We found >50% of M1 neurons had load-related activity for both limbs. Contralateral loads evoked changes in activity ~10ms sooner than ipsilateral loads. We also found corresponding population activities were distinct, with contralateral activity residing in a subspace that was orthogonal to the ipsilateral activity. Thus, neural responses for the contralateral limb can be extracted without interference from the activity for the ipsilateral limb, and vice versa. Our results show that M1 activity unrelated to downstream motor targets can be segregated from activity related to the downstream motor output.
DOI: https://doi.org/10.7554/eLife.48190.001

*For correspondence:
13kc18@queensu.ca

†These authors contributed equally to this work

## Introduction

Several lines of research demonstrate that primary motor cortex (M1) is principally involved in controlling movements of the contralateral side of the body. Anatomically, greater than 90% of corticospinal projections project to the contralateral spinal cord (*Dum and Strick, 1996*; *Brösamle and Schwab, 1997*; *Lacroix et al., 2004*; *Rosenzweig et al., 2009*). Most of the 10% that project ipsilaterally bifurcate and synapse bilaterally, with few thought to synapse purely onto ipsilateral targets (*Rosenzweig et al., 2009*). While there are many monosynaptic projections from M1 to contralateral limb motoneurons (*Bennett and Lemon, 1996*; *Bennett and Lemon, 1996*; *McKiernan et al., 1998*; *Smith and Fetz, 2009*), there are no monosynaptic projections from M1 to ipsilateral motoneurons (*Soteropoulos et al., 2011*). Stimulation in M1 largely produces contralateral movements and occasionally bilateral movements (*Montgomery et al., 2013*). Lastly, inactivation of ipsilateral motor cortex has minimal effects on ipsilateral hand movements (*Nishimura et al., 2007*).

An early study suggested M1 was largely insensitive to ipsilateral movement (*Tanji et al., 1988*). However, several subsequent studies highlight substantial M1 activity during ipsilateral movements (*Donchin et al., 1998*; *Kermadi et al., 1998*; *Cisek et al., 2003*; *Bundy and Leuthardt, 2019*; *Willett et al., 2019*). *Donchin et al. (2002)* examined macaque M1 activity in relation to ipsilateral and contralateral reaching movements. They found 46% of neurons responded to both hand movements, whereas 34% of neurons responded to only contralateral movements and 19% responded to

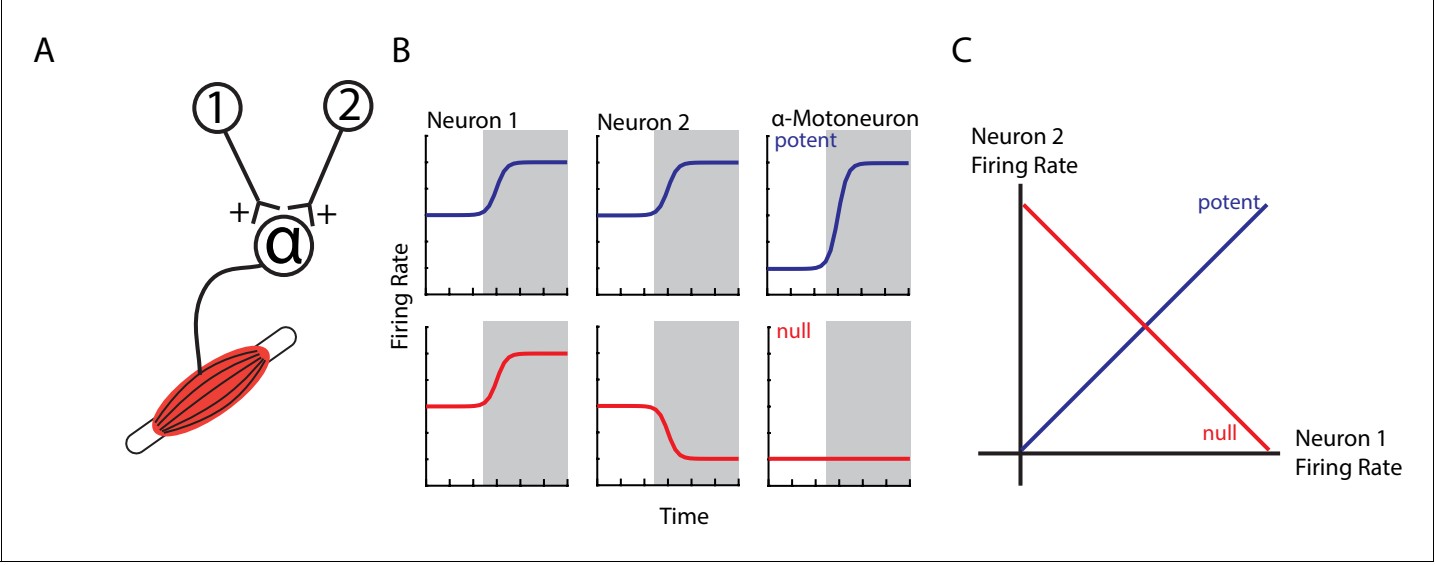

**Figure 1.** Example of orthogonal subspaces. (**A**) Simple example of two neurons that synapse onto an alpha-motoneuron with equal, positive weights. (**B**) Top row, when both neurons increase their firing rate to an event (shaded area), their total activity is summed by the alpha-motoneuron leading to an increase in motor output (potent activity). Bottom row, when neuron one increases and neuron two decreases their firing rate, their net effect cancels at the alpha-motoneuron, thus preventing changes in motor output (null activity). (**C**) State space plots of the firing rates of neurons 1 and 2. Plotting the potent (blue axis) and null (red axis) activity reveals that these patterns are orthogonal with respect to each other (i.e. 90 degrees). Thus, the one-dimensional potent axis represents an orthogonal dimension to the one-dimensional null dimension.
DOI: https://doi.org/10.7554/eLife.48190.002

only ipsilateral movements. As well, functional magnetic resonance imaging (fMRI) revealed considerable motor cortical activity related to ipsilateral movements (*Cramer et al., 1999*; *Kobayashi et al., 2003*; *Gallivan et al., 2011*). In particular, *Diedrichsen et al. (2013)* found areas in motor cortex associated with an ipsilateral digit overlapped with its representation for the corresponding contralateral digit.

An obvious question arises as to why ipsilateral activity in M1 does not cause contralateral limb movement. One hypothesis is that this M1 activity cancels out at spinal circuits (*Druckmann and Chklovskii, 2012*; *Kaufman et al., 2014*). For example, consider two neurons that have equal, excitatory synapses onto an alpha-motoneuron (*Figure 1A*). If both neurons increase their firing rate, the result will lead to excitation in the alpha-motoneuron and thus an increase in motor output (potent pattern, *Figure 1B* top row). Conversely, if one neuron increases and the other neuron decreases its firing rate equally, the net effect on the alpha-motoneuron will be zero leading to no change in motor output (null pattern, bottom row). Examination of the potent and null patterns in state space, where each axis represents the firing rate of a neuron, reveals that these two patterns are in dimensions that are at 90° (orthogonal) to each other (*Figure 1C*). This strategy can allow motor cortex to perform computations necessary for planning an upcoming movement without causing movement (*Churchland et al., 2012*; *Kaufman et al., 2014*; *Elsayed et al., 2016*). Similarly, rapid visual feedback of the hand may be isolated in subspaces orthogonal to the subspaces used for the subsequent motor correction (*Stavisky et al., 2017*). Thus, this hypothesis predicts that ipsilateral activity occupies subspaces that are orthogonal to contralateral activity.

Several studies have shown M1 responds to proprioceptive feedback (*Evarts and Tanji, 1976*; *Wolpaw, 1980*; *Chapman et al., 1984*; *Pruszynski et al., 2011*; *Takei et al., 2018*) and represents loads applied to the contralateral limb (*Cabel et al., 2001*; *Herter et al., 2009*; *Omrani et al., 2014*; *Pruszynski et al., 2014*). Here, we used a postural perturbation task to explore M1 responses to ipsilateral and contralateral motor function. We found ~55% of neurons were active when loads were applied to the contralateral and ipsilateral limbs. However, contralateral loads tended to evoke neural responses that were twice as large as responses for ipsilateral loads. Furthermore, contralateral loads evoked changes in neural activity ~10 ms earlier than ipsilateral loads. Lastly, we found contralateral activity occupied subspaces that were orthogonal to the ipsilateral activity suggesting a

mechanism on how motor cortex can sequester the ipsilateral activity without causing contralateral movement.

## Results

### Kinematics and EMG

Monkeys were trained to keep their right or left hand in a target. Following an unloaded hold period, an unexpected mechanical load was applied to the shoulder and/or elbow joint that moved the limb and required the monkey to return their hand to the target for reward. *Figure 2A-B* display Monkey P's hand motion following mechanical loads applied to the contralateral limb only (right limb). The load caused the perturbed hand to deviate ~1 cm before the monkey returned the hand to the target. In the unperturbed limb (left limb), we observed much smaller hand motion and changes to the speed appeared to start in <100 ms (*Figure 2—figure supplement 1A*). When loads were applied to the ipsilateral limb only, the perturbed limb deviated ~2 cm from the start position while the unperturbed limb showed minimal hand motion (*Figure 2C–D*, supplementary 3B). Note that the hand paths in *Figure 2A and C* are plotted on the same scale.

We compared the integrated hand speed of the contralateral and ipsilateral hand for each context (contralateral versus ipsilateral loads). Hand speeds were an order of magnitude larger when loads were directly applied to the limb versus loads applied to the opposite limb (ratio between perturbed and unperturbed motion: right limb Monkey p=15, Monkey M = 9.7; left limb: Monkey p=10, Monkey M = 8.3). Similar results were found when we quantified maximum hand speed and limb motion for different time epochs (data not shown).

For Monkey P we recorded intramuscular activity from extensors and flexors of the shoulder and elbow joint. *Figure 2E* shows the group averaged muscle response when loads perturbed the limb (blue trace) and when loads perturbed the opposite limb (red trace). A clear increase in muscle activity can be observed in <50 ms when the limb was perturbed. In contrast, there was little change in muscle activity when the opposite limb was perturbed. *Figure 2F* shows the root-mean-squared (RMS) muscle activity during the baseline and perturbation epoch. A significant increase from baseline was detected when the limb was perturbed (paired t-test $t(4)$ = 11.7 p<0.001), while no significant change was detected when the opposite limb was perturbed ($t(4)$ = 0.8 p=0.48). *Figure 2G* compares the change in muscle activity from baseline in the perturbation epoch when the limb was perturbed (contralateral loads) and when the opposite limb was perturbed (ipsilateral loads). Most samples lie near the abscissa indicative of a larger response when the limb was perturbed than when the opposite limb was perturbed. We applied a two-way ANOVA with epoch (levels: baseline, perturbation) and load direction (levels: eight load combinations) as factors to the muscle activity. When the limb was perturbed, four of five muscle samples had a significant interaction effect between epoch and load combination ($F(7,176)=5.3$, p<0.001, $F(7,160)=15.4$ p<0.001, $F(7,96)=11.4$, p<0.001, $F(7,96)=2.3$ p=0.04) while the remaining muscle exhibited a significant main effect of epoch ($F(1,160)$, p=0.006). No muscle samples were significant when the opposite limb was perturbed.

### Neural recordings

From Monkeys P and M, we recorded 92 and 130 neurons from M1, respectively. We first determined if each neuron was sensitive to the loads by applying a three-way ANOVA with epoch (levels: baseline and perturbation epochs), context (levels: contralateral and ipsilateral) and load direction (eight levels: each load combination) as factors. For Monkey P/M we found 90/91 neurons had either a significant main effect of epoch or interaction effects with epoch, indicating that they were load sensitive (thus referred to as 'load-sensitive' neurons). For the load-sensitive neurons, we determined each neuron's preferred load by a planar fit to the neuron's activity during the perturbation epoch (first 300 ms after load onset). *Figure 3A* shows the activity of an example neuron with a significant fit for the ipsilateral and contralateral loads. This neuron's preferred load was pure shoulder extensor for the ipsilateral loads (left panel) and was shoulder flexor combined with elbow extensor for the contralateral loads (right panel). Note that the firing rate scales are different between the ipsilateral and contralateral activity. *Figure 3B C* show two neurons with a significant fit for contralateral only (B) and ipsilateral only (C) loads, respectively. Note in *Figure 3B* for the ipsilateral loads that this neuron tended to excite for shoulder-flexion and elbow-extension torques, however, this neuron did

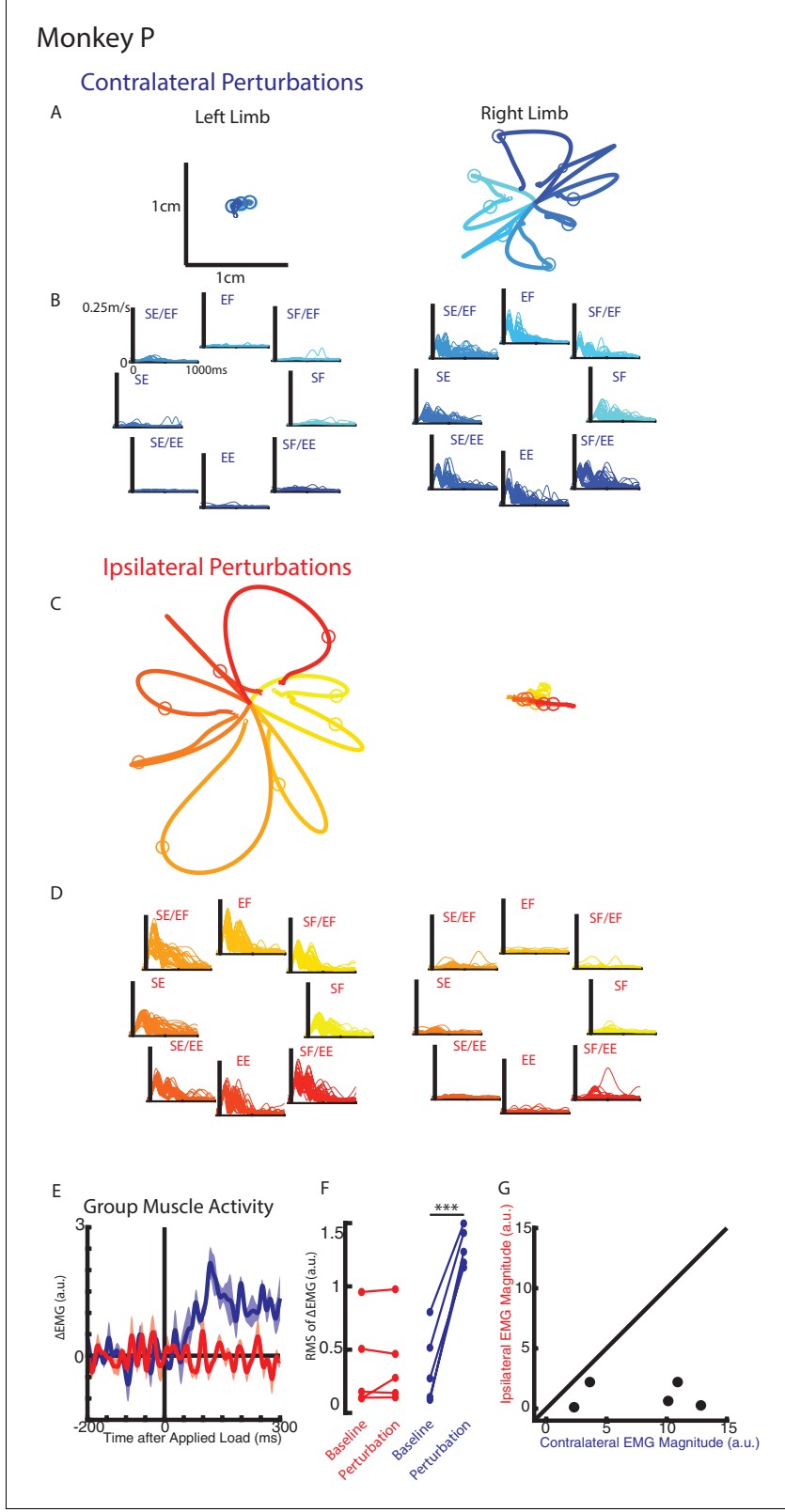

**Figure 2.** Hand kinematics when loads were applied to the contralateral or ipsilateral limbs. (A) Average hand paths for the left and right hand from Monkey P when loads were applied to the contralateral limb (right limb). Circles indicate hand position 300 ms after the loads were applied. Left and right limb data are plotted on the same scale. (B) Single trial (thin lines) and average (thick) hand speeds when contralateral loads were applied. *Figure 2 continued on next page*

*Figure 2 continued*

Black vertical line marks the onset of the load. Speed scales are the same for each load combination and for both hands. (**C–D**) Same as A-B for ipsilateral loads. Note, data are plotted on the same scale as in A-B. See ***Figure 2— figure supplement 1*** showing the unperturbed hand speeds on smaller spatial and temporal scales. (**E**) Group average change in muscle activity recorded from the right limb of Monkey P for contralateral and ipsilateral loads. Note, for presentation purposes, EMG traces were low-pass filtered with a 3^{rd}-order Butterworth filter with a cut-off frequency of 50 Hz. (**F**) Comparison of the root-mean-squared (RMS) muscle activity during the baseline and perturbation epoch. (**E**) Comparison of the average muscle activity for each sample in the perturbation epoch for contralateral and ipsilateral loads. ***p<0.001.
DOI: https://doi.org/10.7554/eLife.48190.003
The following figure supplement is available for figure 2:

**Figure supplement 1.** Kinematic motion of the unperturbed limb.
DOI: https://doi.org/10.7554/eLife.48190.004

not exhibit broad tuning (middle panel) and was thus classified as responsive for contralateral loads only. Four additional neurons are included in *Figure 3—figure supplement 1*.

During the perturbation epoch, for Monkey P/M we found 48/60% of load-sensitive neurons had significant fits for the ipsilateral and contralateral loads, whereas 11/11% of neurons had a significant fit for ipsilateral loads only and 39/18% of neurons had a significant fit for contralateral loads only. *Figure 4—figure supplement 1A,B* compares the probability values for the ipsilateral and contralateral fits. We found no significant correlation for Monkey P (Pearson's correlation coefficient r = −0.1) and a moderate positive correlation for Monkey M (r = 0.4).

During the steady-state epoch (last 1000 ms of trial), for Monkey P/M we found 51/53% of neurons had significant fits for both contexts, whereas 16/13% of neurons had a significant fit for ipsilateral loads only and 30/26% of neurons had a significant fit for contralateral loads only. *Figure 4— figure supplement 1C,D* shows no correlation between probability values for the ipsilateral and contralateral fits for either monkey (Monkey P/M: r = −0.1/0.02).

*Figure 4A,F* shows how the ipsilateral preferred loads are distributed in joint-torque space across all load-sensitive neurons. For Monkeys P and A, we found significant bimodal distributions in both epochs (bootstrap from uniform null distribution: perturbation epoch Monkey P: p<0.001; Monkey M: p<0.001, steady-state Monkey P: p<0.01; Monkey M: p<0.01). For both monkeys, the major axis of the bimodal distributions resided in the quadrants for combined shoulder extension/flexion and elbow flexion/extension (*Figure 4A,F* quadrants 2 and 4, red line). Calculating the difference between a neuron's preferred load during the perturbation and steady-state epochs revealed a significant unimodal distribution with a major axis near zero (*Figure 4D,I*, bootstrap from uniform null distribution, p<0.001, both monkeys).

For the contralateral loads, we found significant bimodal distributions for the perturbation and steady-state (*Figure 4B,G*: perturbation epoch Monkey P: p<0.05; Monkey M: p<0.001, steady-state Monkey P: p<0.01; Monkey M: p<0.05). For Monkey P, the major axes of the bimodal distributions were near the elbow-load axes with a bias towards the quadrants for combined shoulder extension/ flexion and elbow flexion/extension (blue line). For Monkey M we found the bimodal distributions were skewed towards combined shoulder extension/flexion and elbow flexion/extension. Calculating the difference between a neuron's preferred load during the perturbation and steady-state epochs also revealed significant unimodal distributions with a major axis near zero for both monkeys (*Figure 4E,J*, p<0.001).

Next, we examined the difference between each neuron's preferred load between the ipsilateral and contralateral limb. We included all load-sensitive neurons for this analysis. For Monkey P, we found the distribution showed no unimodal structure for either epoch (*Figure 4C*, perturbation epoch and steady-state, p=0.2). For Monkey M we found a unimodal distribution with a major axis near 180° for both epochs (*Figure 4H*, perturbation and steady-state epochs, p<0.01). However, these distributions were qualitatively more dispersed than the distributions that compared changes across epochs. These results suggest that directional tuning is more similar across epochs than across the limbs.

Although neural activity in M1 was observed for both ipsilateral and contralateral motor tasks, most neurons displayed greater activity for the contralateral limb. *Figure 5A E* compares the firing

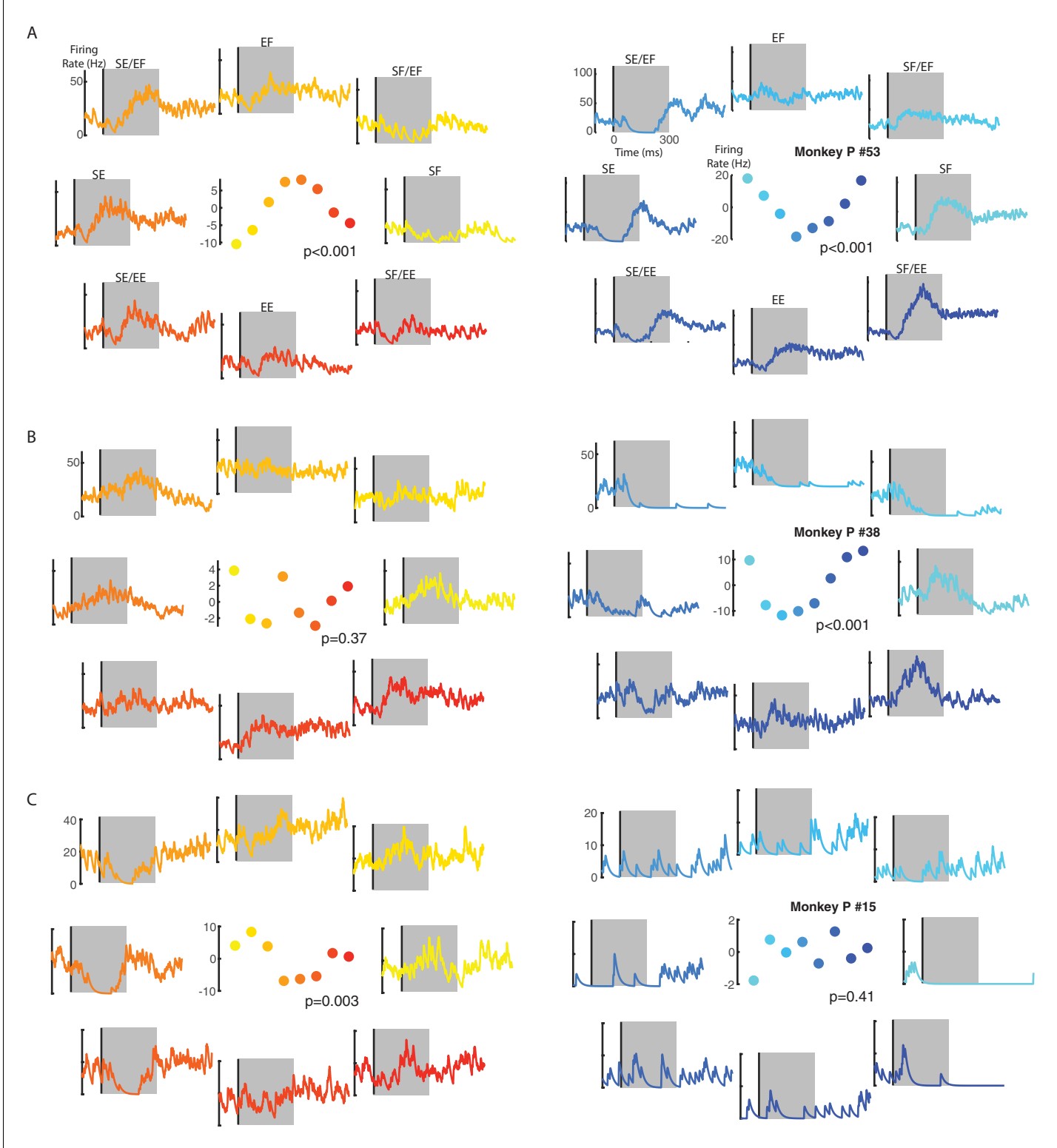

**Figure 3.** Example neuron responses for ipsilateral and contralateral loads. (**A**) A neuron with significant fits for the ipsilateral (left) and contralateral loads (right). Vertical line in each panel denotes the time when the load was applied to the limb. Eight load combinations were applied to each limb and are displayed following the same format in *Figure 10*. Note, the firing rate scales are different between the ipsilateral and contralateral activity. Center panel shows the average firing rate in the perturbation epoch (grey regions) for each load combination. p denotes the probability values for the planar fits. (**B**) A neuron with a significant planar fit for contralateral loads only. (**C**) A neuron with a significant planar fit for ipsilateral loads only. See *Figure 3—figure supplement 1* for four additional example neurons.

*Figure 3 continued on next page*

*Figure 3 continued*

DOI: https://doi.org/10.7554/eLife.48190.005
The following figure supplement is available for figure 3:
**Figure supplement 1.** Additional example neurons.
DOI: https://doi.org/10.7554/eLife.48190.006

rates of each neuron for their preferred load during the perturbation epoch from Monkeys P and M, respectively. Only neurons with significant tuning for at least one of the contexts during the perturbation epoch were included. More than 70% of the neurons had a larger firing rate for contralateral loads (median of all neurons Monkey P: 39 Hz/Nm; Monkey M: 24 Hz/Nm) than ipsilateral loads (Monkey P: 16 Hz/Nm; Monkey M: 13 Hz/Nm). The difference between each neuron's contralateral and ipsilateral firing rate yielded a distribution that was significantly shifted to the right for both monkeys (*Figure 5B F*, Wilcoxon signed-rank test Monkey P: z = 5.1 p<0.001, Monkey M: z = 5 p<0.001) indicating contralateral responses were larger than ipsilateral responses. Similarly, during the steady-state epoch, we found the contralateral loads evoked larger firing rates (*Figure 5C G*, note scales are smaller than for 6A and E; Monkey P: 25 Hz/Nm, Monkey M: 16 Hz/Nm) than ipsilateral loads (Monkey P: 10 Hz/Nm, Monkey M: 10 Hz/Nm). The difference between contralateral and ipsilateral magnitudes yield a distribution that was significantly shifted to the right for both monkeys (*Figure 5D H*, Wilcoxon signed-rank test Monkey P: z = 5.9 p<0.001, Monkey M: z = 4.6 p<0.001). However, approximately a quarter of the neurons displayed larger responses for the ipsilateral limb across epochs and monkeys.

Next, we examined when a change in firing rate could be detected following an applied load. *Figure 6A E* display the average change in firing rate across the population of load-sensitive neurons for Monkeys P and M, respectively. For Monkey P, responses for the contralateral and ipsilateral loads started at 28 ms and 39 ms, respectively. For Monkey M, the response for the contralateral loads started later than for Monkey P at 51 ms. However, the response for the ipsilateral loads was still detected later at 57 ms. *Figure 6B F* compare neurons with onsets for the contralateral and ipsilateral loads. From this population more than 70% of neurons had an onset earlier for contralateral (median onset Monkey P: 68 ms, Monkey M: 79 ms) than ipsilateral loads (Monkey P: 81 ms, Monkey M: 117 ms). The difference between contralateral and ipsilateral onsets yielded a distribution that was significantly shifted to the left for both monkeys (Wilcoxon signed-rank test Monkey P: z = 2.7 p<0.01, Monkey M: z = 4 p<0.001), indicating contralateral responses tended to be earlier than ipsilateral responses. However, 30% of neurons had response times that were earlier for the ipsilateral limb.

As previously mentioned, contralateral loads evoked greater activity than ipsilateral loads, which may have biased the onset earlier for the contralateral activity. We investigated this by first establishing if there was a relationship between activity magnitude and onset within a context. For both monkeys, we found no significant correlations between magnitude and onset timing for the contralateral (Pearson's correlation coefficient: Monkey P: r = −0.08, Monkey M: r = 0.13) and ipsilateral activity (Monkey P: r = 0.12, Monkey M: r = 0.004). Next, we compared the difference in onsets with the ratio of the activity magnitudes between the contralateral and ipsilateral activity (*Figure 6D,H*). In both monkeys, we saw no apparent relationship between the magnitude and onsets (Monkey P/M: −0.08 /- 0.07). We also analyzed a subset of neurons with an absolute magnitude difference that was less than 20 Hz. This subset of neurons had an equal number of neurons with responses that were larger for the contralateral than ipsilateral limb and responses that were larger for the ipsilateral than contralateral limb (*Figure 6—figure supplement 1A,D*). The comparison of the contralateral and ipsilateral onsets still revealed that contralateral loads tended to evoke earlier responses than ipsilateral loads (*Figure 6—figure supplement 1B,E*). For Monkey M, we found a significant leftward shift in the distribution of onset differences (*Figure 6—figure supplement 1F*, Wilcoxon signed-rank test: z = 2.8 p<0.01). For Monkey P we also found a leftward shift in the distribution, however it was not significant (*Figure 6—figure supplement 1C*, z = 1.4 p=0.17). These data suggest that the earlier onset times for the contralateral loads were not simply due to larger changes in activity.

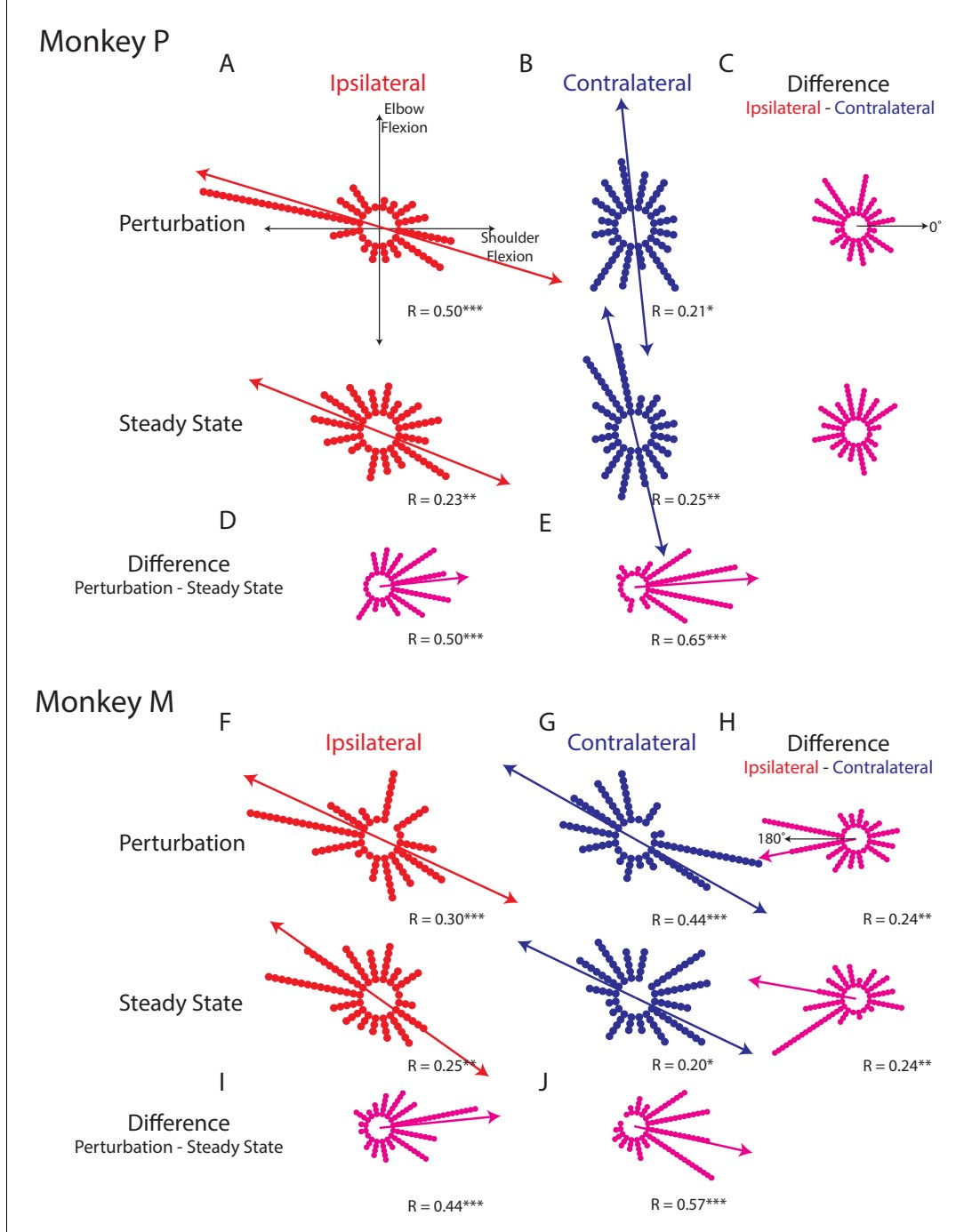

**Figure 4.** Tuning distributions in joint-torque space. (**A**) Polar histograms showing the distribution of tuning curves plotted in joint-torque space for the perturbation (top) and steady-state (bottom) epoch for the ipsilateral loads. R reports the Rayleigh statistic. Red arrow shows the major axis of the bimodal distribution. Only load-sensitive neurons were included. (**B**) Same as A for the contralateral loads. (**C**) Polar histograms showing the change in tuning between the contralateral and ipsilateral loads. Neurons with no change in tuning between contralateral and ipsilateral loads would lie along the 0° axis (top). (**D**) Polar histogram showing the change in tuning between the perturbation and steady-state epochs for ipsilateral loads. Magenta arrows indicate major axis for the unimodal distribution. (**E**) Same as D for contralateral loads. (**F–J**) Same as A-E for Monkey M. *p<0.05, **p<0.01, ***p<0.001.
DOI: https://doi.org/10.7554/eLife.48190.007

The following figure supplement is available for figure 4:

**Figure supplement 1.** Comparison between contralateral and ipsilateral probability values.
DOI: https://doi.org/10.7554/eLife.48190.008

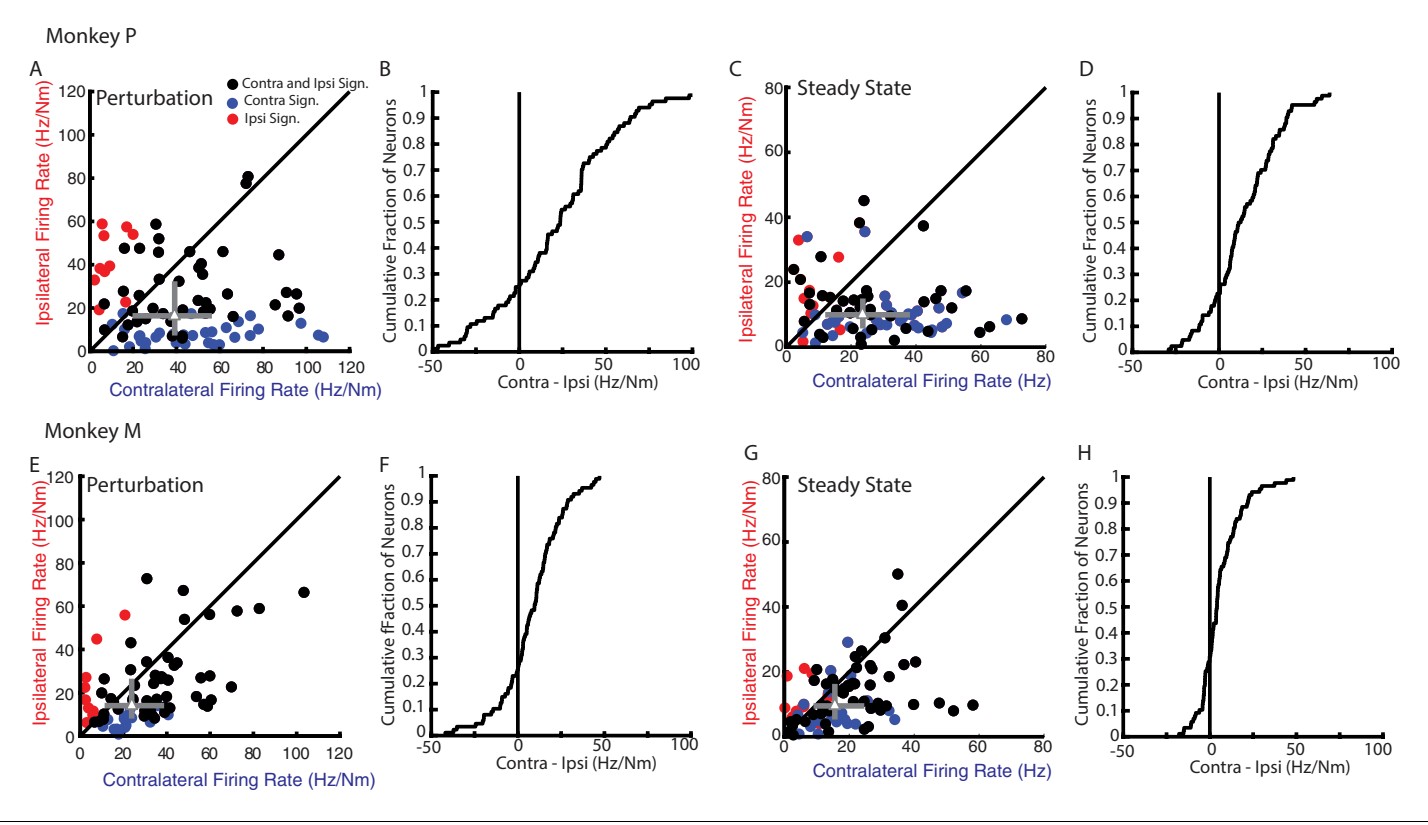

**Figure 5.** Magnitude comparison between contralateral and ipsilateral neural responses. (A) Comparison between the contralateral and ipsilateral firing rates during the perturbation epoch for each neuron as determined by their planar fit. Black circles are neurons with significant fits for contralateral and ipsilateral loads. Blue and red circles are neurons with a significant fit for contralateral or ipsilateral loads only, respectively. Grey triangle and bars represent the median and interquartile range, respectively. (B) The cumulative distribution generated from the difference between contralateral and ipsilateral firing rates. Only neurons with a significant fit for at least one of the contexts were included. (C–D) Same as A-B for steady state. (E–H) Same as A-D for Monkey M.

DOI: https://doi.org/10.7554/eLife.48190.009

## Single electrode analyses

We also ensured our main findings were observed using conventional single electrodes by sampling 34 neurons in Monkey P from the hemisphere that was opposite to the implanted array. We found only 53% of recorded neurons were load-sensitive, a lower percentage than we observed for the opposite hemisphere. The population response started at 35 ms for the contralateral loads and 47 ms for the ipsilateral loads (data not shown). We also found the magnitude of the contralateral response was larger than the ipsilateral response for 70% of neurons in the perturbation epoch (median magnitude: contralateral 19.6 Hz/Nm, ipsilateral 9.2 Hz/Nm). Finally, during the posture epoch we found ~50% of neurons with a larger contralateral response (contralateral 6.4 Hz/Nm, ipsilateral 4.1 Hz/Nm).

## Control analyses

One explanation for the substantial change in tuning between the contralateral and ipsilateral loads is that we included neurons with significant fits for only one context. We analyzed a subset of neurons with significant fits for both the contralateral and ipsilateral loads. We again found no unimodal structure for Monkey P (perturbation: R = 0.19 p=0.23, steady-state: R = 0.17, p<0.29), whereas for Monkey M we found a moderate unimodal distribution during the perturbation (R = 0.55, p<0.001) and steady-state epochs (R = 0.37, p<0.001). For comparison we examined the change in tuning across epochs using neurons with significant fits for both the perturbation and steady-state epochs. For both monkeys, we found strong unimodal distributions for the contralateral (Monkey P/M:

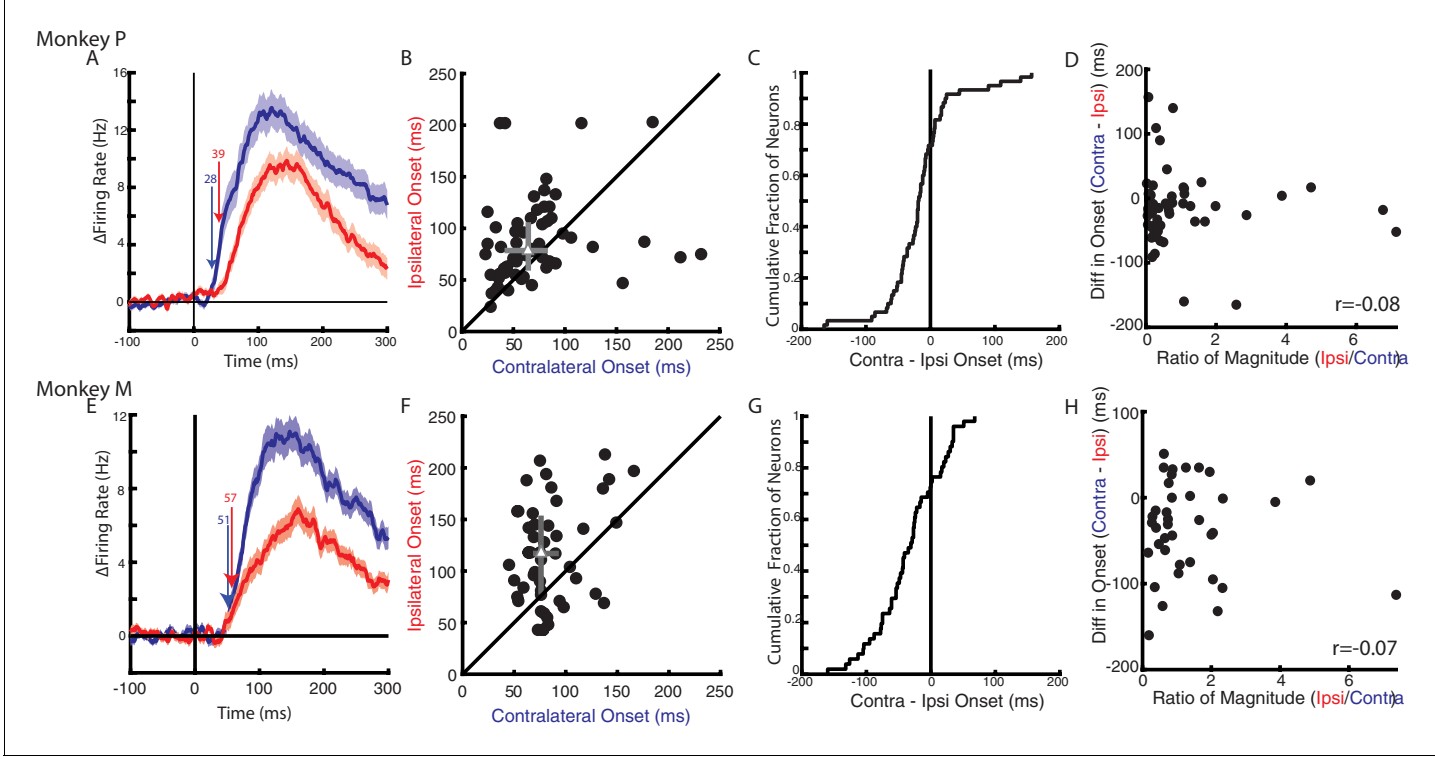

**Figure 6.** Timing of neural responses for the contralateral and ipsilateral loads. **(A)** The average change in firing rate across the population for the contralateral and ipsilateral loads. Arrows mark the onset when a significant change in baseline was detected for contralateral (blue) and ipsilateral (red) loads. Mean and SEM are plotted. All load-sensitive neurons were included. **(B)** Comparison of the onsets for the contralateral and ipsilateral loads. Grey triangle and bars represent the median and interquartile range, respectively. **(C)** The cumulative distribution generated from the difference between contralateral and ipsilateral onset times. **(D)** Comparison between onset differences between the contralateral and ipsilateral activity with the ratio of their activity magnitudes. r denotes Pearson's correlation coefficient. **(E–H)** Same as A-D for Monkey M. **(B–D, F–H)** Only neurons that were load-sensitive and had significant onsets for both contexts were included.

DOI: https://doi.org/10.7554/eLife.48190.010

The following figure supplement is available for figure 6:

**Figure supplement 1.** Onset timing while controlling for magnitude effects.
DOI: https://doi.org/10.7554/eLife.48190.011

R = 0.77/0.79, p<0.001) and ipsilateral loads (Monkey P/M: R = 0.77/0.79, p<0.001). This analysis supports our earlier conclusion that tuning is more similar across epochs than between limbs.

We also analyzed a subset of neurons that minimized any overlap caused by pooling neurons across recording sessions ('non-overlapping' neurons, see Materials and methods) and found essentially the same results. This subset contained 60 and 104 neurons for Monkeys P and M, respectively. We found the change in tuning between the contralateral and ipsilateral limb was not significantly unimodal for Monkey P and weakly unimodal for Monkey M (*Supplementary file 1*). Contralateral magnitudes were significantly larger than ipsilateral magnitudes in the perturbation epoch (*Supplementary file 2*; Wilcoxon signed-rank test Monkey P/M: z = 4/4.2 p<0.001) and steady-state epoch (Wilcoxon signed-rank test Monkey P/M: z = 4.9/3.4 p<0.001). Lastly, we found 70% of neurons had earlier onsets for the contralateral loads than ipsilateral loads (*Supplementary file 3*). The difference between contralateral and ipsilateral onsets yielded a distribution that was significantly shifted to the left for both monkeys (Wilcoxon signed-rank test Monkey P/M: z = 2.3/3.3 p=0.02/0.0008).

## Population analysis

Next, we examined how correlations between neurons change for the contralateral and ipsilateral loads. As suggested by *Figure 1B–C*, if contralateral and ipsilateral activity resided in separate

subspaces, then the correlation between pairs of neurons should also change between these two contexts. We focus on the perturbation epoch exclusively given the low firing rate of ipsilateral activity in the steady-state epoch. *Figure 7A C* compares the correlation coefficients between all neuron pairs for the contralateral and ipsilateral loads. The resulting distributions are dispersed, indicating that the correlations during the contralateral loads were not predictive of their correlations during the ipsilateral loads. *Figure 7B D* show the distribution of the absolute change in correlation coefficients between all neuron pairs for the ipsilateral and contralateral loads (black trace). For Monkeys P and M, we found the median change in correlation coefficient to be 0.32 and 0.20, respectively. These median changes were significantly larger than a null distribution generated by comparing pairwise correlations across randomly separated trials within a context (*Figure 7B D*, red and blue traces, p<0.001 see Materials and methods).

The substantial change in correlation structure across the population suggest ipsilateral and contralateral activity resided in separate, orthogonal subspaces. These subspaces were identified using principal component analysis (PCA) which finds a linear weighting of each neuron's response that

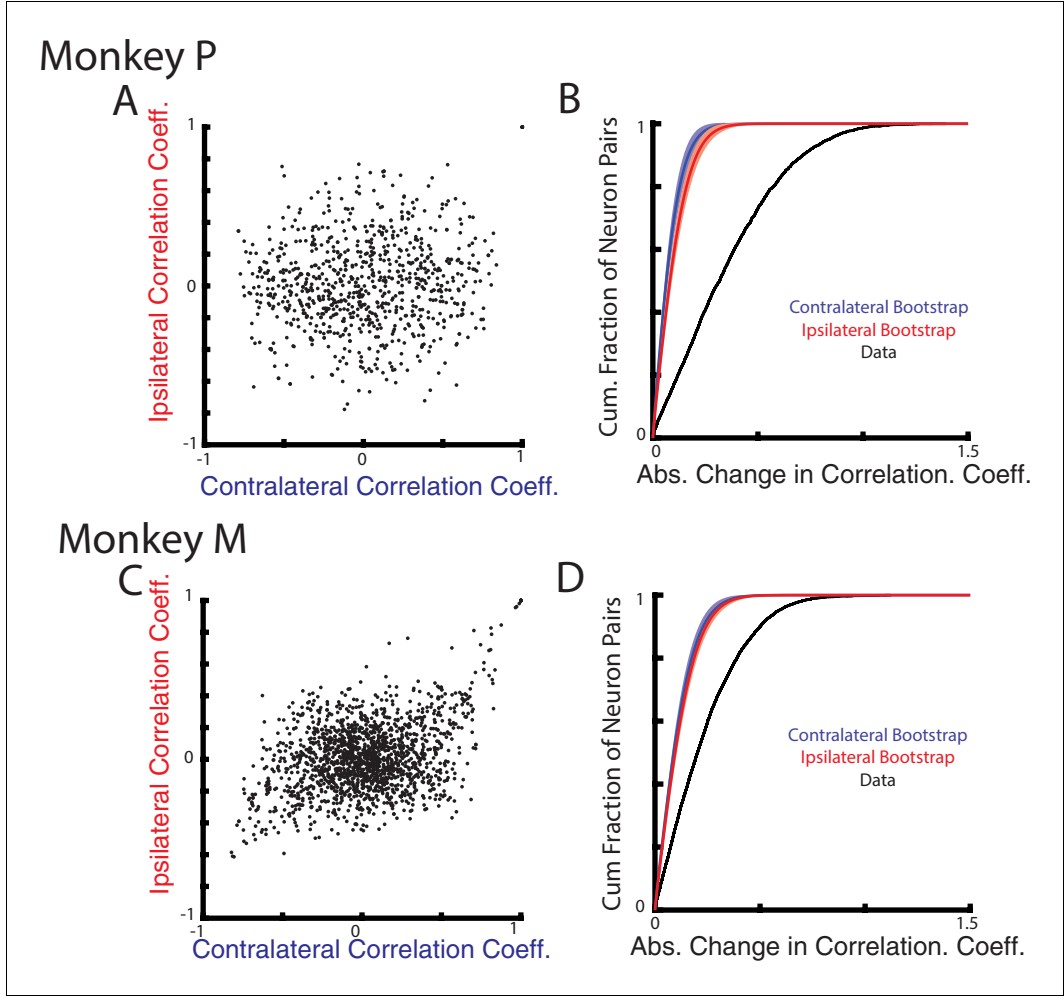

**Figure 7.** Correlation changes between neurons for ipsilateral and contralateral loads. (**A**) Comparison of the pairwise correlation coefficients between neurons during the ipsilateral and contralateral loads. (**B**) The cumulative sum of the absolute change in the pairwise correlation coefficient between the ipsilateral and contralateral loads (black). A bootstrap distribution was generated by separating ipsilateral trials into two distinct groups and calculating the absolute change in pairwise correlation coefficient (red trace). This was repeated 1000x. Shaded region shows three standard deviations from the mean. This was repeated for the contralateral data (blue). (**C–D**) Same as A-B for Monkey M.

DOI: https://doi.org/10.7554/eLife.48190.012

captures the largest amount of variance. **Figure 8A D** show the variance captured by the top-ten principle components generated from the ipsilateral activity for Monkeys P and M, respectively. For Monkey P/M, these components captured 74/61% of the ipslateral variance, while accounting for ~20% of the contralateral variance. Likewise, the top-ten principle components generated from the contralateral activity captured 81/65% of the contralateral variance but only captured <20% of the ipsilateral variance (**Figure 8B,E**). We computed the alignment index to quantify how orthogonal the top ipsilateral and contralateral principle components were (**Elsayed et al., 2016**). The alignment index ranges from 0, indicating perfect orthogonality, to one indicating perfect alignment. For Monkey P/M, we found the average alignment index to be 0.19/0.29, respectively (**Figure 8C,F**). These values were significantly smaller than the alignment indices generated by randomly sampling from

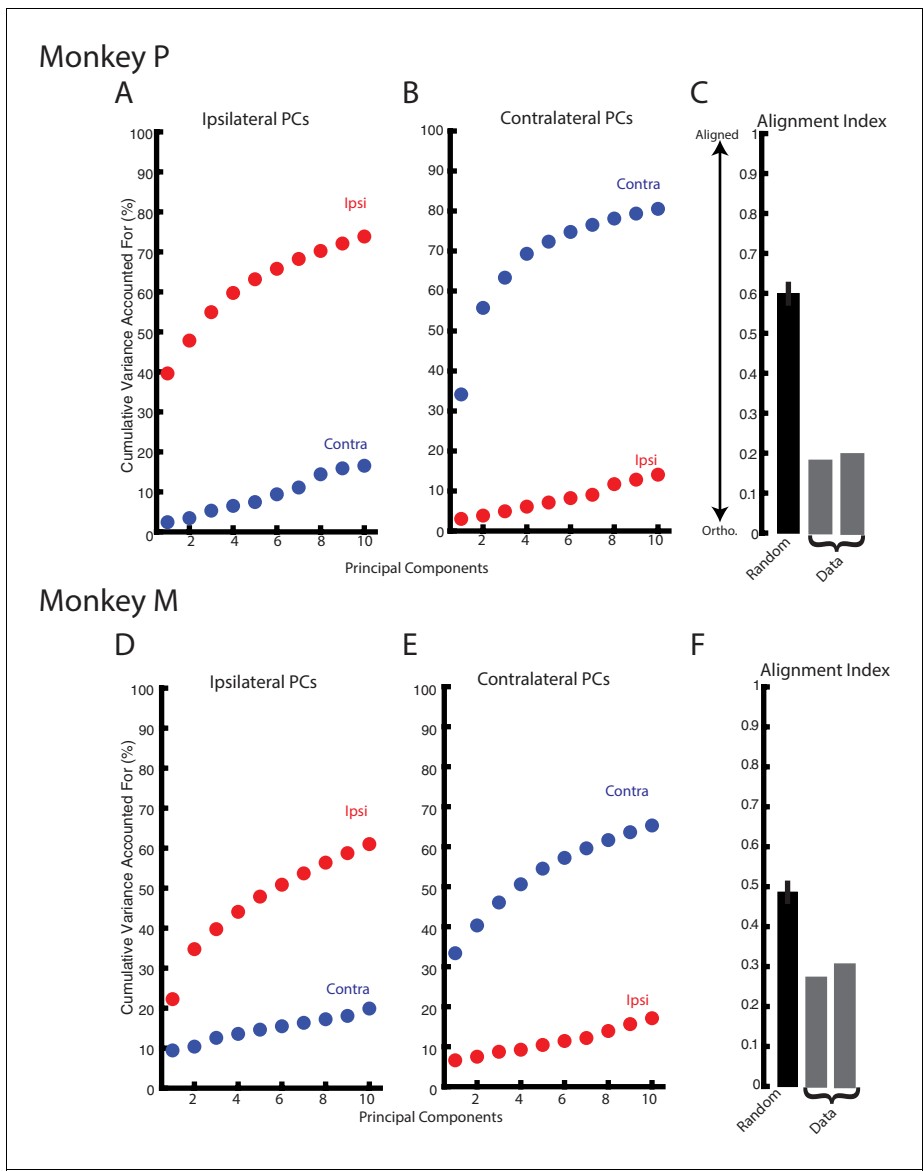

**Figure 8.** Ipsilateral and contralateral activity reside in nearly orthogonal subspaces. (A) The cumulative variance explained of the ipsilateral and contralateral activity from the ten largest ipsilateral principle components. (B) The cumulative variance explained of the ipsilateral and contralateral activity from the ten largest contralateral principle components. (C) The alignment index generated by randomly sampling from the data covariance matrix (black, mean + standard deviation) and the two indices generated from the ipsilateral and contralateral principle components (grey). (D–F) Same as A-C for Monkey M.
DOI: https://doi.org/10.7554/eLife.48190.013

the data covariance matrix (mean: Monkey P 0.60; Monkey M 0.49; bootstrap p<0.001), indicating ipsilateral and contralateral activity are more orthogonal than expected by random chance. Examining the subset of non-overlapping neurons also yielded alignment indices (Monkey P/M: 0.25/0.29) that were significantly smaller than alignment indices generated by randomly sampling from the data covariance matrix (p<0.001).

We found the top-ten principle components captured less variance than observed in previous studies of motor cortex (*Elsayed et al., 2016*; *Miri et al., 2017*). This may reflect the kernel we used to estimate firing rates as it is not as smooth as the gaussian kernels used previously. For comparison with the literature, we re-analyzed our data after convolving with a gaussian kernel (standard deviation 20 ms). We found the top-ten ipsilateral principle components captured 90/77% of the ipsilateral variance for Monkey P/M respectively, while capturing <25% of the contralateral activity. Similarly, the top-ten contralateral principle components captured 92/80% of the contralateral variance for Monkey P/M, while capturing <20% of the ipsilateral variance. The average alignment index for Monkey P/M was 0.16/0.27 and was significantly smaller than indices generated by randomly sampling from the data covariance matrix (mean: Monkey P 0.67; Monkey M 0.55; bootstrap p<0.001).

One possibility for the separation between contralateral and ipsilateral activity is that PCA identified two separate groups of neurons for each context. This seems unlikely given that we observed a substantial proportion of neurons that were active during both contexts. Nonetheless, we addressed this issue following a similar procedure to *Perich et al. (2018)*. We summed the absolute value of the weights from the top-ten ipsilateral ($w_{ipsi}$) and contralateral ($w_{contra}$) principal components. We then calculated the ipsilateral and contralateral difference divided by their sum $\left(w_{ipsi} - w_{contra}\right)/\left(w_{ipsi} + w_{contra}\right)$. The histogram generated from all neurons did not appear bimodal (data not shown), as would be expected if PCA was identifying separate groups of neurons. Instead we observed a unimodal distribution that was not significantly different from a normal distribution (Kolmogorov-Smirnov test: Monkey P: D(92)=0.06 p=0.9, Monkey M: D(130)=0.08 p=0.3). These data indicate PCA was not simply isolating separate neural populations for ipsilateral and contralateral activity.

Although PCA identified subspaces for ipsilateral and contralateral activity that were close to orthogonal, they were still partially aligned (particularly for Monkey M). We identified subspaces that capture the ipsilateral and contralateral activity that were also orthogonal with respect to each other by using a joint optimization procedure from *Elsayed et al. (2016)*. For Monkey p/M, we found three ipsilateral dimensions that captured 54/37% of the ipsilateral variance, while only capturing 3%/4% of the contralateral variance. Projecting the ipsilateral activity onto the three ipsilateral dimensions revealed substantial time-varying dynamics (*Figure 9A C*, left column). By comparison, projecting the contralateral activity onto the ipsilateral dimensions revealed little change from baseline (right column). Comparing the ipsilateral and contralateral activity in the ipsilateral dimensions revealed an average difference between activities of 105%/116% for Monkey P/M, which was significant (*Equation 4*, Materials and methods; bootstrap p<0.001).

Similarly, the three contralateral dimensions we found captured 62/44% of the contralateral variance for Monkey P/M, while only capturing 3%/4% of the ipsilateral variance. Examining the contralateral activity in the three contralateral dimensions revealed substantial time-varying dynamics (*Figure 9B C* right columns) while ipsilateral activity in these dimensions changed little from baseline (left columns). Comparing the contralateral and ipsilateral activity in the contralateral dimensions revealed an average difference between activities of 99%/104% for Monkey P/M, which was significant (bootstrap p<0.001). These time-series demonstrate that linearly summing each neuron's response can isolate ipsilateral neural activity from the contralateral activity, and vice versa.

## Discussion

We examined how load-related activity for the ipsilateral and contralateral limbs are represented using a postural perturbation task. The contralateral responses tended to be larger and earlier postperturbation than the ipsilateral responses. However, a substantial proportion of neurons responded to loads applied to the ipsilateral limb, and in some cases, these responses were larger and occurred earlier than for the contralateral limb. We also demonstrated that each neuron's preferred load for the ipsilateral limb tended to be unrelated to its preferred load for the contralateral limb. Lastly, the

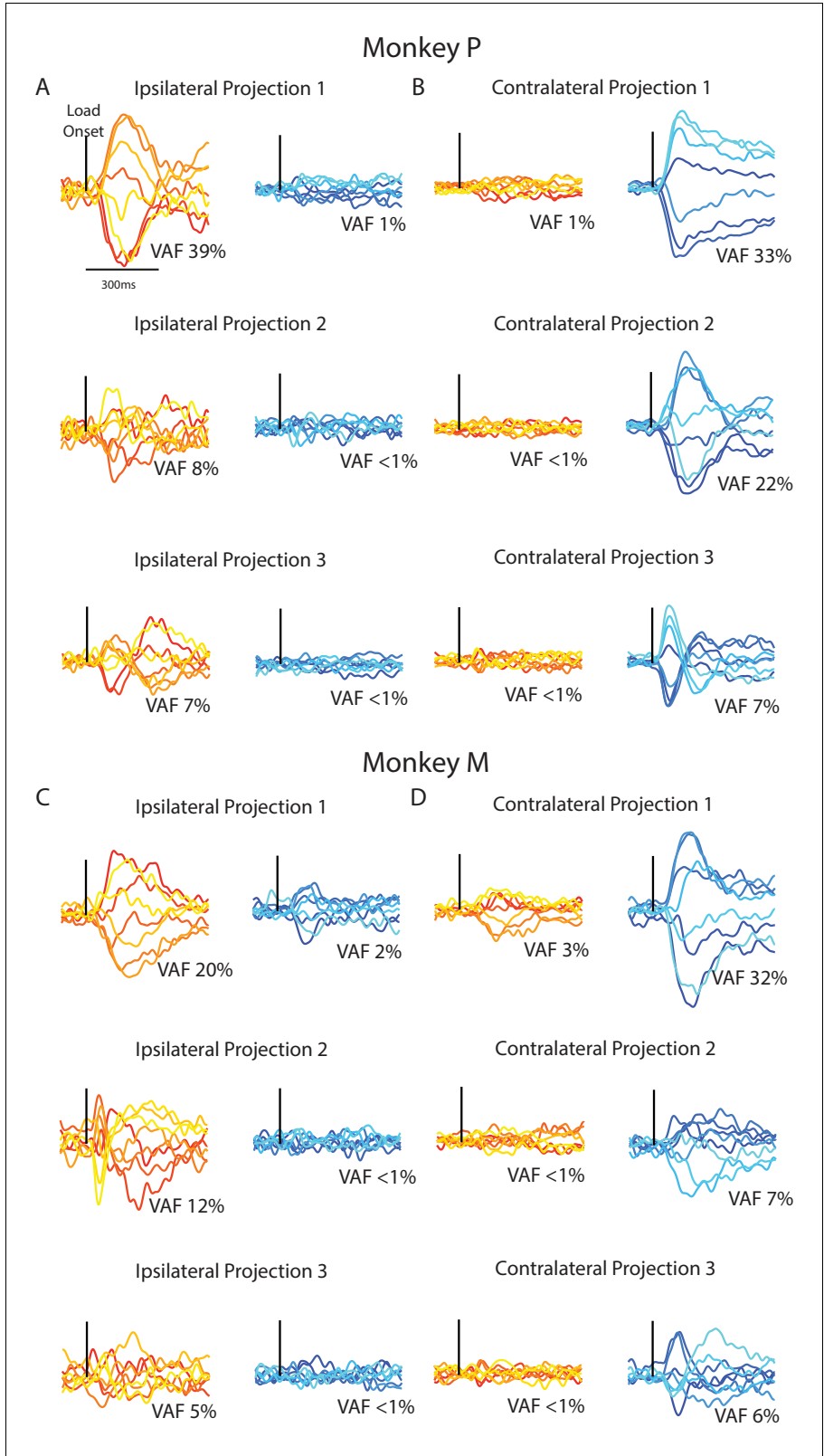

**Figure 9.** Time series from the ipsilateral and contralateral dimensions. (A) For Monkey P, the time series generated by projecting the ipsilateral (left) and contralateral (right) activity onto the three ipsilateral. Black line indicates when the load was applied. (B) Same as A for the three contralateral dimensions. (C–D) Same A-B for Monkey M.

DOI: https://doi.org/10.7554/eLife.48190.014

ipsilateral and contralateral activities occupied orthogonal subspaces, allowing the ipsilateral and contralateral responses to be separated by a weighted sum of each neuron's response.

We found ~55% of load-sensitive neurons in M1 responded to ipsilateral and contralateral disturbances similar to reaching studies (*Donchin et al., 1998*; *Donchin et al., 2001*; *Donchin et al., 2002*; *Donchin et al., 2002*; *Kermadi et al., 1998*; *Cardoso de Oliveira et al., 2001*; *Gribova et al., 2002*; *Cisek et al., 2003*). The fastest response to an ipsilateral load was ~10 ms after the fastest response for a contralateral load, suggesting the corpus callosum was relaying the ipsilateral information (*Aboitiz et al., 1992*; *Ringo et al., 1994*; *Fendrich et al., 2004*; *Caminiti et al., 2013*). This delay also matches the delay in task-dependent feedback responses between the two limbs during bimanual motor tasks (*Marsden et al., 1981*; *Mutha and Sainburg, 2009*; *Dimitriou et al., 2012*; *Omrani et al., 2013*).

Following a perturbation, we observed a small amount of motion in the unperturbed limb starting in <100 ms. This motion may be the result of energy transferred through the trunk from the perturbed limb to the unperturbed limb. The exoskeleton supports the forearm and upper arm which helps dampen motion transfer between the arms, however some transfer of motion is likely inevitable since the arms are mechanically coupled by the trunk. Alternatively, given the onset timing, the motion may have been generated by small muscular contractions in the unperturbed limb. However, we failed to observe any changes in muscle activity in the unperturbed limb.

This small motion in the unperturbed limb is an unlikely explanation for why we observed M1 activity when ipsilateral loads were applied. Previously, we showed M1 responses are less sensitive to applied loads if the limb is not engaged in a task (*Omrani et al., 2014*; *Omrani et al., 2016*). In the current study, we demonstrated for ipsilateral loads that M1 activity was approximately half the size of the activity for contralateral loads. Correspondingly, we showed for ipsilateral loads that the kinematic motion of the contralateral (unperturbed) hand was approximately ten times smaller than for contralateral loads. Thus, M1's sensitivity to contralateral hand motion would have to increase substantially when not engaged in the task, inconsistent with *Omrani et al. (2014)* and *Omrani et al. (2016)*. Lastly, during the steady-state epoch when hand motion had ceased, we still observed significant neural activity for ipsilateral loads. Thus, most M1 activity during the ipsilateral loads was likely related to the ipsilateral limb.

The preferred load of M1 neurons were not uniformly distributed for the contralateral limb in joint-torque space. Instead, M1 neurons were bimodally distributed with higher proportion of neurons maximally active for combined elbow flexion and shoulder extension, and combined elbow extension and shoulder flexion, consistent with previous studies (*Cabel et al., 2001*; *Herter et al., 2009*). A similar distribution of preferred loads exists for proximal limb muscles (*Kurtzer et al., 2006*). Notably, the bimodal distribution is opposite to the anatomical action of the bi-articular muscles, reflecting that mono-articular muscles shift their preferred load to consider loads at the non-spanned joint (*Nozaki et al., 2005*). A neural network trained to control a two-joint limb displays a similar distribution in preferred loads across the network when bi-articular muscles are included (*Lillicrap and Scott, 2013*).

In a similar manner, perturbation activity in M1 related to the ipsilateral limb also reflects features related to motor output. When the ipsilateral loads were applied, the distribution of preferred loads had a similar bimodal distribution in joint-torque space. Recently, *Ames and Churchland (2019)* compared M1 responses to contralateral and ipsilateral cycling movements and found M1 activity could reliably decode ipsilateral muscle activity. Thus, ipsilateral-related activity is not an abstract representation of ipsilateral motor function. Rather, it reflects the mechanical and anatomical properties of the ipsilateral limb even though the descending projections predominantly target the contralateral musculature.

The present study highlights a reorganization of the population response for the ipsilateral and contralateral limbs. First, the preferred load was largely unrelated between the contralateral and ipsilateral limbs. *Cisek et al. (2003)* examined the relationship between a neuron's preferred direction during reaches with the contralateral and ipsilateral limb. They found the preferred directions were similar for rostral areas in motor cortex between contralateral and ipsilateral reaches, whereas caudal areas exhibited little relationship. For Monkey P, the recorded neurons exhibited no relationship between the preferred loads for the contralateral and ipsilateral limb, whereas for Monkey M we found a small correlation. This may reflect that Monkey M's array was placed ~2 mm more rostral than for Monkey P (data not shown).

Relatedly, *Steinberg et al. (2002)* constructed population vectors (PV) from M1 activity during contralateral reaches by linearly weighting each neuron's response. Applying the same weights to the ipsilateral activity found vector magnitudes that were negligible. It seems likely we would have observed a similar reduction in PV magnitude given the substantial change in tuning, although the bimodal distribution of preferred-loads would have a complex effect on PV direction and magnitude (*Scott et al., 2001*).

Importantly, we demonstrate a mechanism by which ipsilateral activity in M1 does not lead to contralateral motor output. We found M1 activity related to the two limbs was not simply random or uncorrelated. Rather, ipsilateral activity was specifically organized into a subspace that was almost orthogonal to the subspace of the contralateral limb, similar to *Ames and Churchland (2019)*. This allowed us to isolate the contralateral activity from the ipsilateral activity simply by summing a linear weighting of each neuron's response (*Kaufman et al., 2014*; *Elsayed et al., 2016*). This may provide a mechanism for how ipsilateral activity does not generate motor output. Thus, the synaptic weights of M1 neurons that project onto spinal circuits could be assigned such that the net effect of ipsilateral activity cancels out.

However, it is important to note that one could not infer that contralateral and ipsilateral activity reside in orthogonal subspaces based solely on the change in preferred load direction (*Kaufman et al., 2014*; *Elsayed et al., 2016*). By describing only the preferred loads, we ignore the rich, temporal dynamics of M1 neurons (*Churchland and Shenoy, 2007*). These temporal dynamics can arise on time scales shorter than the epoch used to assess the preferred load thus contributing little to a neuron's preferred load (ex. *Figure 9A* projection three contralateral activity). If these temporal dynamics dominate the variance of a neuron's firing rate, than one can have a substantial change in preferred loads without a substantial change in the subspace the population resides in. Both analyses provide complementary evidence for a network reorganization for representing the contralateral and ipsilateral limb.

It is interesting to note that the subspaces related to the contralateral limb may not be fixed. M1 activity remains in the same subspace during motor learning (*Golub et al., 2018*; *Perich et al., 2018*; *Vyas et al., 2018*), and during reaching tasks with different load conditions (*Gribble and Scott, 2002*; *Gallego et al., 2018*) and initiation cues (*Lara et al., 2018*). However, *Miri et al. (2017)* demonstrated that activity during locomotion and reaching occupied orthogonal subspaces that may allow motor cortex to engage separate spinal circuits for each behavior. A similar mechanism for engaging spinal circuits may exist for load-related activity during posture and reaching. Previously, we found a neuron's gain during posture showed no relationship with its gain during reaching (*Kurtzer et al., 2005*), which may reflect neural activity in separate subspaces for these motor actions. This suggests that different classes of behavior, posture versus movement versus locomotion, have distinct subspaces. This may reflect differences in control policies for each behavior, and how motor responses to sensory feedback must be modified across these behavours (*Todorov and Jordan, 2002*; *Scott, 2016*).

Brain regions that utilize the ipsilateral activity can also weight each neuron's response to extract the ipsilateral activity without interference from the contralateral activity. *Li et al. (2016)* highlights how activity in ipsilateral motor cortex improves the robustness of activity in the contralateral motor cortex. The ipsilateral subspace may also allow for flexible bimanual coupling when both limbs are utilized for a bimanual task (*Marsden et al., 1981*; *Diedrichsen, 2007*; *Mutha and Sainburg, 2009*; *Dimitriou et al., 2012*; *Omrani et al., 2013*; *Córdova Bulens et al., 2018*). In addition, this subspace may be altered following spinal cord injuries or stroke to allow ipsilateral M1 to contribute to descending motor output (*Cramer et al., 1997*; *Nishimura et al., 2007*).

The ability to isolate neural activity in subspaces that do not influence spinal circuits has implications for a long-standing debate regarding M1's involvement in descending control. Many studies have found M1 activity correlates with muscle activity (*Evarts, 1968*; *Humphrey, 1972*; *Murphy et al., 1985*; *Bennett and Lemon, 1996*; *Scott, 1997*; *Sergio and Kalaska, 1998*; *Kakei et al., 1999*; *Cherian et al., 2013*; *Oby et al., 2013*; *Heming et al., 2016*), suggesting that M1 activity reflects low-level features of the motor output. However, other studies have correlated M1 activity with whole limb movements (*Georgopoulos et al., 1982*), trajectories (*Schwartz, 1992*; *Schwartz, 1993*) and other high level parameters of motion (*Johnson et al., 1999*). Even when high and low-level features of motor action are dissociated, some activity still appears to be related to high level features of the movement (*Thach, 1978*; *Sergio and Kalaska, 1998*; *Kakei et al., 1999*;

*Russo et al., 2018*). Our data suggest that both sets of information could be present in M1 simultaneously, but only low-level motor commands influence spinal circuits (*Lalazar et al., 2019*). By retaining different types and sources of information in different subspaces, neural activity can both control motor output and process other types of information simultaneously whether it is related to high level features of movement or even the ipsilateral limb.

## Materials and methods

### Animal and apparatus

Studies were approved by the Queen's University Research Ethics Board and Animal Care Committee. Two non-human primates (*Macaca mulatta*) sat in a primate chair and were trained to place their arms into exoskeleton robots that were attached to the primate chair (Kinarm, Kingston, Canada; *Scott, 1999*). Each exoskeleton includes two troughs that support the forearm/hand and upper arm segments, and the linkage lengths were adjusted to align the robot's joints with the monkey's shoulder and elbow joints.

The animals performed a postural perturbation task similar to our previous studies (*Herter et al., 2009*; *Pruszynski et al., 2014*). On each trial, the monkey maintained its right or left hand, represented by a white cursor (0.5 cm diameter), at a stationary virtual target (0.8 cm diameter, red for right hand, blue for left hand, luminance matched). These targets were placed approximately in front of each respective shoulder (*Figure 10A*). Only one target was shown at a time, thus only one arm was used in a trial. The arm that was not used in a trial could rest in the environment, unrestrained by the robot in the horizontal plane. After an initial unloaded hold period of 500–1000 ms, a random flexion or extension step load was applied to the shoulder and/or elbow joint. After the load was applied, the monkey had 1000 ms to return their hand to the target. The monkey had to maintain its hand inside the target for 1000–1500 ms to receive a water reward.

Eight load conditions were used for each limb, consisting of torques that caused elbow flexion (EF), elbow extension (EE), shoulder flexion (SF) or shoulder extension (SE), or the four multi-joint combinations of torques (SF/EF, SF/EE, SE/EF, SE/EE, *Figure 10B*). The magnitude of the torques were 0.20 Nm for single joint loads, and 0.14 Nm at each joint for multi-joint torques. One block included all torque conditions interleaved for the contralateral and ipsilateral limbs (16 total trials). A minimum of 10 blocks were completed in a single session.

### Neural, EMG, and kinematic recordings

After training, monkeys underwent surgery to implant 96-channel Utah Arrays (Blackrock Microsystems, Salt Lake City, UT) into the arm region of M1. Surgeries were performed under aseptic conditions and a head fixation post was also attached to the skull using dental cement. A dura substitute (GORE PRECLUDE Dura Substitute, W.L.Gore and Associates Inc) was placed over the array before the dura was re-attached (GOR-TEX Suture, W.L.Gore and Associates Inc). Spike waveforms were sampled at 30 kHz and acquired using a 128-Channel Neural Signal Processor (Blackrock Microsystems, Salt Lake City, UT). For Monkey P we also implanted a chamber above the right arm area of M1 and neurons were recorded using single electrodes over 18 recording sessions (*Herter et al., 2009*). For Monkey P's array, we recorded neurons from three behavioral sessions spaced 4 months apart. For Monkey M, we also collected from three behavioral sessions, with over a year between the first and second session, and three months between the second and third session.

Spikes were manually sorted offline (Offline Sorter, Plexon Inc, Dallas TX) using a space spanned by the top two principal components and the peak-to-trough voltage difference. Only well isolated single units were used for analysis and no outliers were removed. For our main analysis, we combined all the neurons across all sessions for a given Monkey, thus treating all neurons as independent samples. We felt this was justified given the length of time between recordings. Previous studies estimate that 10–50% of neurons can be reliably tracked over a two-week period (*Jackson and Fetz, 2007*; *Tolias et al., 2007*; *Dickey et al., 2009*; *Fraser and Schwartz, 2012*). Furthermore, *Fraser and Schwartz (2012)* tracked neurons over the course of 100 days and found that virtually no neurons could be reliably tracked over this extended time period. The number of reliably-tracked neurons also appears to decay exponentially over time (*Tolias et al., 2007*; *Fraser and Schwartz, 2012*). For our study, the smallest time period between behavioral sessions was three months. Given

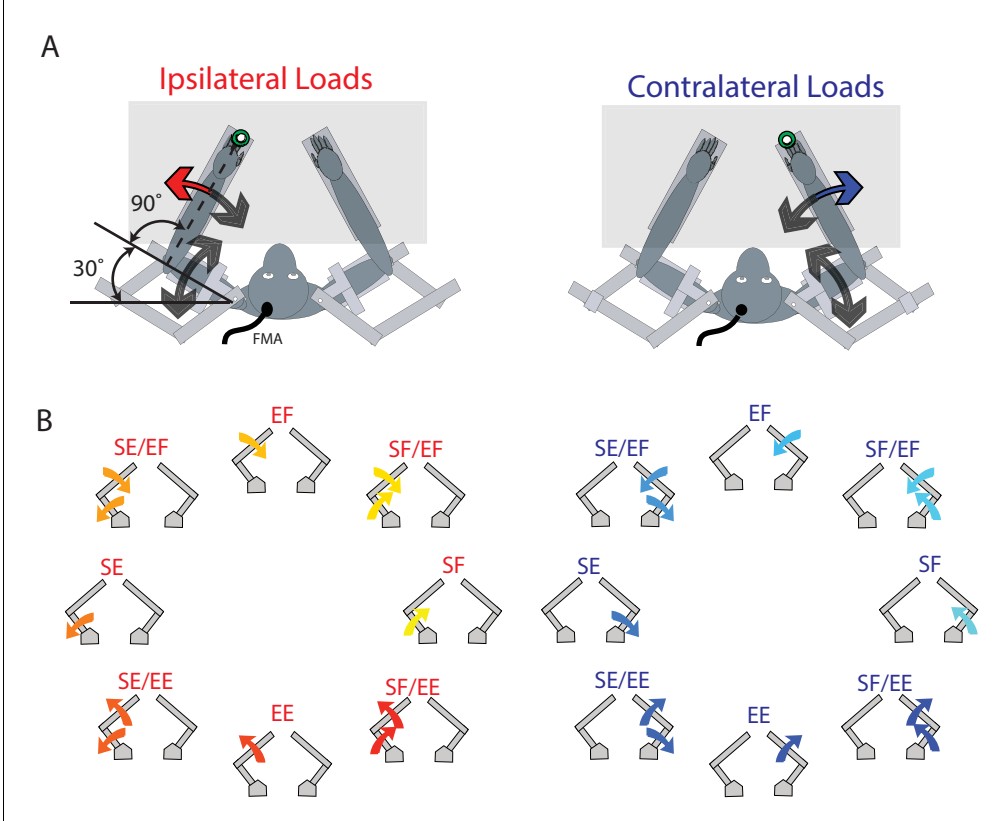

**Figure 10.** Experimental set-up. (**A**) The monkey's left and right arms were supported by the exoskeleton. Monkeys were trained to return their hand to the goal target (green) when mechanical loads were applied to the left (ipsilateral; red arrow) and right (contralateral; blue arrow) hand. Visual feedback of their hand was presented as a white cursor. The target was placed such that to reach the target monkeys required to flex their shoulder 30° and place their elbow at 90°. (**B**) For ipsilateral and contralateral loads, combinations of flexion and extensions torques were applied to the shoulder and elbow joints. Arrows show the net applied torque for each load combination. Abbreviations: SF shoulder flexion, SE shoulder extension, EF elbow flexion, EE elbow extension, FMA floating micro-electrode array.

DOI: https://doi.org/10.7554/eLife.48190.015

an exponential decay in reliably-tracked neurons and a decay half-life of 50% per 2 weeks, this amounts to an overlap of neurons between sessions of ~2%. However, to further minimize the possibility that neurons were overlapping, we analyzed a subset of neurons. This subset was constructed by comparing adjacent behavioral sessions. If a neuron was recorded on the same array channel during both sessions, the neuron recorded on the later behavioral session was discarded. The only exception was for Monkey M between the first and second session, as more than a year had passed making it highly unlikely that any neurons were overlapping.

We recorded intramuscular electromyographic (EMG) activity from Monkey P by inserting two thin wires into the muscle belly of the brachioradialis, lateral and long head of triceps, long head of the biceps and pectoralis major (*Scott and Kalaska, 1997*). Signals were digitized at 1 kHz and recorded by the Neural Signal Processor.

Joint angles, velocities, and accelerations for both arms, were recorded at 1 kHz by the Neural Signal Processor. All offline analysis was performed using custom MATLAB scripts (The MathWorks, Inc, Natick, Massachusetts, United States) and can be accessed at https://github.com/KevinCross/ContralpsiCode (*Cross, 2019*; copy archived at https://github.com/elifesciences-publications/ContralpsiCode).

## Data and statistical analysis

*Kinematic analysis.* Kinematic signals were filtered using a 3rd order low-pass Butterworth filter with a cutoff frequency of 10 Hz. Integrated hand speed was calculated by summing the hand speed from 0 to 300 ms after the perturbation.

## EMG analysis

EMG signals were aligned to the perturbation onset and filtered using a 6th order bandpass Butterworth filter with a frequency range of 20–200 Hz. Signals were then rectified and further smoothed with a 6th order low-pass Butterworth filter with a cutoff frequency of 100 Hz. EMG activity was averaged in two epochs: baseline which was the last 200 ms before the load was applied, and a perturbation epoch which included the first 300 ms after the load was applied. We tested for load sensitivity by applying a two-way ANOVA with epoch (levels: baseline and perturbation) and load combination (levels: eight load combinations) as factors. Samples were deemed significant if a main effect of epoch or an interaction effect was significant ($p<0.05$). Group average signals were generated by finding the muscle's preferred load direction and normalizing the EMG signal by the mean activity in the perturbation epoch.

## Spike trains smoothing and epochs

Spikes were convolved with an asymmetric kernel approximating a post-spike potential (1 ms rise time and 20 ms fall time) (*Thompson et al., 1996*). This kernel is causal, meaning it only influences the firing rate after rather than before the spike. This prevents backward biasing when estimating a neuron's onset that would arise with non-causal Gaussian kernels. We constructed three trial-averaged histograms, one that was aligned to the perturbation onset and spanned the first 200 ms before the load onset (baseline epoch), the first 300 ms after the perturbation (perturbation epoch), and one that spanned the last 1000 ms of the trial (steady-state epoch).

## ANOVA analysis

For each neuron's smoothed firing rates, we applied a three-way ANOVA with epoch (levels: baseline and perturbation), context (levels: contralateral and ipsilateral) and load combination (levels: all eight load combinations) as factors. We classified neurons as load sensitive if they had a significant main effect for epoch, or any interaction effects with epoch ($p<0.05$).

## Preferred load direction

For each neuron we averaged the smoothed firing rates within each epoch and subtracted off the mean signal. We then fit a planar model (MATLAB *regress*) that predicted a neuron's firing rate from the applied loads. From these fits we calculated a neuron's preferred load and its corresponding activity magnitude. Rayleigh unimodal and bimodal statistics (R-statistics) were used to assess the distribution of preferred loads across the population (*Batschelet, 1981*; *Lillicrap and Scott, 2013*). We compared our results to a bootstrapped distribution generated by randomly sampling angles from a uniform distribution spanning 0–360°. We matched the number of angles we sampled with the number of neurons in our population of interest. We then calculated the unimodal and bimodal R-statistics for the resulting bootstrap distribution and repeated this procedure 1000 times. Significance was assessed by calculating the percentage of times we found the bootstrapped sample had a larger R-statistic than our neuron population.

## Timing onset

We estimated the onset time for each neuron by first finding the load combination that generated the absolute largest change in firing rate from baseline during the perturbation epoch. We then calculated the trial average from that load combination and also included trials from the two spatially adjacent load combinations to improve the onset estimate (*Herter et al., 2009*). Onsets were calculated by finding the first time point that exceeded baseline activity by three standard deviations and remained above that threshold for 20 consecutive milliseconds (*Omrani et al., 2016*).

## Pairwise Correlations and PCA

We used previously established methods for our correlational and PCA analyses (*Elsayed et al., 2016*; *Miri et al., 2017*). First, we down sampled the smooth firing rate of each neuron by sampling every 10ms. We then soft-normalized the contralateral and ipsilateral activity of each neuron by its maximum firing rate plus 5 spikes/s. For each context (contralateral vs ipsilateral loads), we subtracted the mean activity across the eight perturbation types for each time bin. For contralateral and ipsilateral activity, we constructed matrices C and I $\in R^{NxCT}$ where N is the number of neurons, C the number of mechanical loads (8) and T the number of time points (perturbation epoch: 30 time points after down sampling). Note, all recorded neurons were used, including those that did not exhibit significant load sensitivity.

We compared how the pairwise correlations changed between the two contexts by computing the correlation between each possible pair of neurons during each context. This yielded two correlation coefficients for each pair of neurons, one for each context. We then calculated the absolute difference between the two correlation coefficients for each neuron pair. We compared our results with a bootstrapped null distribution (*Figure 7B D*), as previously described (*Miri et al., 2017*). This distribution is generated from the assumption that the absolute difference in correlation between the two contexts is simply due to averaging over a finite number of trials. For each neuron, we randomly assigned each of its trials during one of the contexts to two separate groups and generated trial-averaged firing rates. Within each group we calculated all pairwise correlation coefficients between neurons and then calculated the absolute difference in coefficients between groups. This was repeated 1000.

PCA was performed on matrices C and I using singular value decomposition. We selected the top-ten principal components for each context and calculated the amount of contralateral and ipsilateral variance each projection captured.

We determined how aligned the top-ten contralateral and ipsilateral principle components were by the alignment index, as described previously (*Elsayed et al., 2016*):

$$A_{C \, on \, I} = \frac{Tr\left(PC_{contra}^T Cov_{ipsi} PC_{contra}\right)}{Tr\left(PC_{ipsi}^T Cov_{ipsi} PC_{ipsi}\right)} \tag{1}$$

$$A_{I \, on \, C} = \frac{Tr\left(PC_{ipsi}^T Cov_{contra} PC_{ipsi}\right)}{Tr\left(PC_{contra}^T Cov_{contra} PC_{contra}\right)} \tag{2}$$

Where $PC_{contra}$ and $PC_{ipsi} \in R^{10xCT}$ are matrices containing the top-ten contralateral and ipsilateral principal components, $Cov_{contra}$ and $Cov_{ipsi}$ are the contralateral and ipsilateral covariance matrices ($R^{nxn}$), and Tr is the trace operator. Equation 1 reflects a ratio of how much variance the top-ten contralateral principal components could explain of the ipsilateral activity with how much the top-ten ipsilateral principal components could explain of the ipsilateral activity (the max variance any ten linear projections could capture). This metric was also computed using the contralateral activity (Equation 2). The alignment index ranges from 0, which indicates the contralateral and ipsilateral principal components are perfectly orthogonal, to 1 which indicates the contralateral and ipsilateral principal components are perfectly aligned.

We assessed if the alignment indices were significant by a bootstrap procedure as described in *Elsayed et al. (2016)*. Initially, PCA was performed on the data matrix generated by concatenating the contralateral and ipsilateral matrices, C and I. Two sets of ten components were randomly sampled from the data matrix with a probability of selection weighted by the amount of variance captured by that component. This procedure was repeated 1000 times to generate a null distribution.

## Orthogonalization

We used a joint optimization technique described by *Elsayed et al. (2016)* that forced dimensions for each respective limb to be orthogonal with respect to each other. We then observed the amount of residual activity related to one limb that was expressed in the other limb's dimensions. The cost function used for this optimization was

$$Q_{contra}, Q_{ipsi} = argmax_{Q_{contra}, Q_{ipsi}} \frac{1}{2} \left( \frac{Tr\left(Q_{contra}^T \, Cov_{contra} \, Q_{contra}\right)}{\sum_{i=1}^{d} \sigma_{contra}(i)} + \frac{Tr\left(Q_{ipsi}^T \, Cov_{ipsi} \, Q_{ipsi}\right)}{\sum_{i=1}^{d} \sigma_{ipsi}(i)} \right) \quad (3)$$

Where d is the number of dimensions, $\sigma_{contra}(i)$ and $\sigma_{ipsi}(i)$ are the i-th singular values of the contralateral and ipsilateral covariance matrices, and $Q_{contra}$, $Q_{ipsi}$ are matrices $\in R^{Nxd}$ composed of the orthonormal components. With the orthonormal constraints on $Q_{contra}$, $Q_{ipsi}$ the problem reduces to an optimization problem on a Steifel manifold which we found using the Manopt toolbox for MAT-LAB (*Cunningham and Ghahramani, 2015*; *Boumal et al., 2014*). The resulting projections $Q_{contra}$, $Q_{ipsi}$ were then each used to reduce the contralateral and ipsilateral activity matrices C and I ($C_{contra}$, $I_{contra}$, and $C_{ipsi}$, $I_{ipsi}$, respectively). For presentation purposes, we first applied PCA before plotting to order the projections from largest to smallest amount of variance accounted for, and we also smoothed the time series with a 3$^{rd}$ order low-pass Butterworth filter with a cut-off frequency of 25Hz. However, all analyses utilized the original, unsmoothed time series.

The relative difference between the contralateral and ipsilateral time series in the orthogonal, contralateral dimensions was calculated as

$$\frac{\|C_{contra} - I_{contra}\|}{\|C_{contra}\|} * 100 \quad (4)$$

Where $C_{contra}$ and $I_{contra}$ are the projections onto the contralateral dimensions for the contralateral and ipsilateral activity, respectively. In order to estimate the variability with this metric, we bootstrapped ipsilateral trials and calculated the relative difference 1000 times. A similar calculation was computed for the ipsilateral projections.

# Acknowledgements

We thank Kim Moore, Simone Appaqaq, Justin Peterson, and Helen Bretzke for their laboratory and technical assistance and members of the LIMB lab for constructive criticisms. This work was supported by grants from the Canadian Institute of Health Research. EAH was supported by an NSERC scholarship. KPC was supported by an OGS scholarship. TT was supported by Uehara memorial foundation. SHS was supported by a GSK chair in Neuroscience.

# Additional information

### Competing interests
Stephen H Scott: Co-founder and chief scientific officer of Kinarm which commercializes the robot used in the study. The other authors declare that no competing interests exist.

### Funding

| Funder | Grant reference number | Author |
| --- | --- | --- |
| Canadian Institutes of Health Research | CIHR MOP 84403 | Stephen H Scott |
| Canadian Institutes of Health Research | CIHR PJT 153445 | Stephen H Scott |

The funders had no role in study design, data collection and interpretation, or the decision to submit the work for publication.

### Author contributions
Ethan A Heming, Conceptualization, Data curation, Formal analysis, Investigation, Visualization, Methodology, Writing—original draft, Writing—review and editing; Kevin P Cross, Formal analysis, Investigation, Visualization, Methodology, Writing—original draft, Writing—review and editing; Tomohiko Takei, Formal analysis, Methodology; Douglas J Cook, Performed surgery for array

implant; Stephen H Scott, Conceptualization, Resources, Supervision, Funding acquisition, Methodology, Writing—original draft, Project administration, Writing—review and editing

### Author ORCIDs
Kevin P Cross ⓘ https://orcid.org/0000-0001-9820-1043
Tomohiko Takei ⓘ http://orcid.org/0000-0002-6429-5798
Stephen H Scott ⓘ http://orcid.org/0000-0002-8821-1843

### Ethics
Animal experimentation: Studies were approved by the Queen's University Research Ethics Board and Animal Care Committee (#Scott-2010-035).

### Decision letter and Author response
Decision letter https://doi.org/10.7554/eLife.48190.023
Author response https://doi.org/10.7554/eLife.48190.024

## Additional files

### Supplementary files
• Supplementary file 1. Comparison of Rayleigh statistic between the non-overlapping subset of neurons and the original neuron population. Significance assessed based on sampling from a uniform distribution. *p<0.05, **p<0.01, ***p<0.001
DOI: https://doi.org/10.7554/eLife.48190.016

• Supplementary file 2. Comparison of firing magnitude between the non-overlapping subset of neurons and the original neuron population.
DOI: https://doi.org/10.7554/eLife.48190.017

• Supplementary file 3. Comparison of onset timing between the non-overlapping subset of neurons and the original neuron population.
DOI: https://doi.org/10.7554/eLife.48190.018

• Transparent reporting form DOI: https://doi.org/10.7554/eLife.48190.019

### Data availability
Neural and kinematic data have been submitted to the Dryad repository and can be accessed at https://doi.org/10.5061/dryad.06nr12f.

The following dataset was generated:

| Author(s) | Year | Dataset title | Dataset URL | Database and Identifier |
|---|---|---|---|---|
| Heming EA, Cross KP, Takei T, Cook DJ, Scott SH | 2019 | Data from: Independent representations of ipsilateral and contralateral arms in primary motor cortex | https://doi.org/10.5061/dryad.06nr12f | Dryad Digital Repository, 10.5061/dryad.06nr12f |

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
