## [Decision Letter]

Thank you for submitting your article "Independent representations of ipsilateral and contralateral arms in primary motor cortex" for consideration by *eLife*. Your article has been reviewed by two peer reviewers, and the evaluation has been overseen by a Reviewing Editor and Richard Ivry as the Senior Editor. The reviewers have opted to remain anonymous.

The reviewers have discussed the reviews with one another and the Reviewing Editor has drafted this decision to help you prepare a revised submission.

Summary:

The paper provides an investigation into the neuronal responses in motor cortex during perturbations to the contralateral and ipsilateral arm. The main findings are that a) M1 represent the perturbations to both arms b) the responses to ipsilateral perturbations a slightly slower than those to contralateral perturbation (but see #2), and c) the population responses are orthogonal, allowing for independent encoding of both perturbations.

Essential revisions:

The reviewers highlighted 3 major areas of concerns that should be addressed:

1) The authors should provide more information about how recordings were combined across days. The Materials and methods state (subsection “Neural, EMG and kinematic recordings”) that neurons were recorded from 3 behavioral sessions spaced 2-3 months apart. Because the electrodes of chronically-implanted Utah arrays are not moved between sessions, there is always a concern that recordings from different days could be picking up an overlapping set of units. How many of the recorded neurons are shared across the recording days? How does this influence the statistical validity of the single neuron and population analyses presented in the paper (some of which are based on the assumption of statistical independence between units)?

2) The analysis of neural onset times is interesting and constitutes a novel major finding of the paper. However, the results could potentially be affected by the fact that contralateral perturbation responses tended to be larger than ipsilateral responses. If a neuron's response has the same time course in two conditions, but reaches a higher firing rate in one condition, then a set threshold for detecting the response time will yield an earlier for the higher firing rate condition. Does the earlier response time for contralateral movements still hold if controlling for the magnitude of firing rate changes? A subset analysis on neurons with matched response magnitudes (or a scatter plot of onset difference vs. response magnitude ratio) may be useful to address this concern.

3) The control analysis presented in Figure 8 should be clarified, revised or removed. One major concern was that the result presented in Figure 8B and C (larger spread of ipsilateral weights as compared to contralateral weights) may be statistical artifact related to the fact that the variances (or sums-of-squares) of the regressors (velocity of the contralateral and ipsilateral arm) are vastly different. The predicted variance of the regression coefficients is lawfully related to inverse of the variance of the regressor. More precisely in the context of multiple regression with design matrix X, the var(b) is proportional to inv(X'*X). Additionally, the ipsilateral movement is correlated to that on the contralateral arm, further increasing the variance of the weights.

Thus arguments of "higher sensitivity" or " context dependent selectivity" need to be carefully reinvestigated.

Overall, the reviewers also commented that the clarity of the writing needs to be improved.

Reviewer #1:

Overall, I found this paper to be well-written, with interesting and important implications for a major motor control mystery: how can motor cortex have neural responses during movements of the ipsilateral arm without evoking contralateral arm movements? The authors suggest that ipsilateral arm related activity resides within muscle null dimensions, thus enabling neural responses to cancel out at the level of downstream circuits, providing no net excitation. They use both single neuron analyses, neuron pair analyses, and state-space analyses to support their conclusion, providing strength and diversity of analyses. They further provide a very detailed analysis of neurons' responses to contralateral and ipsilateral arm perturbations, which helps to ground the paper and relate the conclusions to previous studies comparing contralateral and ipsilateral arm related responses in motor cortex. I have a few major suggestions, shared below, which I envision the authors should be able to fully address without major new experiments or analyses. If the authors address all of these concerns, I would be quite supportive of this paper's publication at *eLife*.

1) The authors should provide more information about how recordings were combined across days. The Materials and methods state (subsection “Neural, EMG and kinematic recordings”) that neurons were recorded from 3 behavioral sessions spaced 2-3 months apart. Because the electrodes of chronically-implanted Utah arrays are not moved between sessions, there is always a concern that recordings from different days could be picking up an overlapping set of units. Were any controls performed to ensure that no neurons were duplicated in the dataset?

2) The analysis of neural onset times could potentially be affected by the fact that contralateral perturbation responses tended to be larger than ipsilateral responses. If a neuron's response has the same time course in two conditions, but reaches a higher firing rate in one condition, then a set threshold for detecting the response time will yield an earlier for the higher firing rate condition. Does the earlier response time for contralateral movements still hold if controlling for the magnitude of firing rate changes?

Reviewer #2:

This manuscript describes a study of the activity in M1 in response to both ipsilateral and contralateral limb movements imposed by an exoskeleton. The title and Abstract are strongly focused on the second half of the study, the idea that the subspaces occupied by the activity driven by the two different arms are independent of one another. The types of methods they employ are being used in an increasing number of studies to investigate a wide range of properties of the networks of neurons that can be recorded by chronically implanted electrode arrays. The first (and lengthier) part of the study uses more traditional methods to analyze the basic properties of single neurons in response to perturbations of the two limbs. I edited the first part rather closely, until I got to the control section. I found the description to be unnecessarily hard to follow, but much more importantly, assuming I understood what they did, it raised major concerns in my mind, rather than alleviating any. Unless these can be adequately explained away, this strikes me as a tremendous amount of description and analysis of signals, the origin of which, is pretty unclear.

Comments related to my major concern:

“contralateral limb motion that was transferred through the body”. Meaning motion of the trunk with the arm stable, that causes shoulder rotation? Seems awfully unlikely. Doesn't the Kinarm constrain trunk motion?

“We addressed this by fitting the average neural activity when contralateral loads were applied to the average contralateral hand velocity in the perturbation epoch (data not shown).” I must not be following this, as I don't follow how this addresses the concern. You're trying to predict M1 from ipsilateral hand motion using a mapping derived from contra hand motion? I can't imagine that working well, or how it serves as a control. I gather the Kinarm was not being used to stabilize the non-moving limb?

“Contralateral and ipsilateral activity evoked a change in firing rate over a similar range for this neuron, but contralateral loads evoked larger hand speeds.” I find this to be really confusing. Activity doesn't "evoke" firing rate; that's how it's measured. Is that what you mean? Furthermore, is the (essentially zero) ipsilateral hand movement that which resulted from (was "evoked by") loads applied to the opposite limb? Unless I'm missing something pretty basic, this section is much more complicated than necessary. The opposite hand simply doesn't move. What more is there than that?

“We fit the evoked neural activity when ipsilateral loads were applied to the corresponding contralateral hand motion.” I confess I'm completely lost. What is meant by, "ipsilateral loads were applied to the corresponding contralateral hand motion." Does it mean mapping from the (tiny) non-moving hand speed to contralateral M1 activity?

“indicating the size of the ipsilateral weights were larger than the corresponding contralateral weights.” In fact, they differ by more than an order of magnitude, presumably because the non-perturbed ("contralateral") hand was moving so slowly. And yet the (tiny) hand movement accompanying loads on the opposite side of the body predicted contralateral M1 well? This causes major concerns about the validity of the entire study, if M1 activity is better predicted by the tiny contralateral hand movements than by the ipsilateral movements (albeit, that generalization was doomed from the outset). I wonder how well you can decode ipsilateral and contralateral limb movement from M1 activity when perturbations drive the ipsilateral arm? If they are of similar quality, or contra decoding is better, it would be a huge concern.

---

## [Author Response]

Essential revisions:The reviewers highlighted 3 major areas of concerns that should be addressed:1) The authors should provide more information about how recordings were combined across days. The Materials and methods state (subsection “Neural, EMG and kinematic recordings”) that neurons were recorded from 3 behavioral sessions spaced 2-3 months apart. Because the electrodes of chronically-implanted Utah arrays are not moved between sessions, there is always a concern that recordings from different days could be picking up an overlapping set of units. How many of the recorded neurons are shared across the recording days? How does this influence the statistical validity of the single neuron and population analyses presented in the paper (some of which are based on the assumption of statistical independence between units)?

We thank the reviewers for raising this concern as overlapping neuron population could have an impact on our results. We incorrectly stated the durations between recordings as 2-3 months. For Monkey P, we included recordings that were collected in April 2014, August 2014 and December 2014. For Monkey M we included recordings collected in January 2017, May 2018 and August 2018. We felt that the length of time between behavioural sessions would have resulted in very few, if any, neurons overlapping between sessions. Previous studies have estimated how well chronically implanted arrays can reliably record from the same neurons across multiple days. These estimates range from 10-50% of neurons could be reliably tracked over a two-week period (Jackson and Fetz, 2007; Tolias et al., 2007; Dickey et al., 2009; Fraser and Schwartz, 2012). Furthermore, (Fraser and Schwartz, 2012) tracked neurons over the course of 100 days and found that virtually no neurons could be reliably tracked over this extended time period. The number of reliably-tracked neurons appear to decay exponentially over time (Tolias et al., 2007; Fraser and Schwartz, 2012). For our study, the smallest time period between behavioural sessions was three months. Given an exponential decay in reliably-tracked neurons with a decay half-life of 50% per 2 weeks, this amounts to an overlap of neurons between sessions of 2%. This explanation has been added to the Materials and methods (subsection “Neural, EMG and kinematic recordings, second paragraph).

However, to ensure our results were not affected by any overlapping neurons, we have now analyzed a subset of neurons in each Monkey. This subset was constructed by comparing adjacent behavioural sessions. If a neuron was recorded on the same array channel during both sessions, the neuron recorded on the later behavioural session was discarded. The only exception was for Monkey M between the first and second session, as more than a year had past making it highly unlikely that any neurons were overlapping. Applying this procedure resulted in 60 neurons for Monkey P and 104 neurons for Monkey M. We then recalculated the metrics that were critical to our study. First, we still found a neuron’s tuning for contralateral and ipsilateral loads were still largely unrelated, and that tuning between the perturbation and steady-state epochs were still quite similar (Supplementary file 1). Second, the firing magnitude of a neuron for contralateral loads was still twice as large as for ipsilateral loads (Supplementary file 3). Third, we detected the activity changes for contralateral loads were earlier than for ipsilateral loads (Supplementary file 2). Lastly, we found the subspaces for contralateral and ipsilateral activity were still nearly orthogonal with respect to each other (Author response table 1). Values in Author response table 1 have been stated in the Results section (subsection “Population Analysis”, second paragraph).

Author response table 1: Comparison of alignment index between the non-overlapping subset of neurons and the original neuron population. Significance assessed based on sampling from the data covariance matrix. ***p<0.001

Non-Overlapping NeuronsOriginal Neuron PopulationAlignment IndexMonkey P: R=0.25*** Monkey M: R=0.29***Monkey P:R=0.19*** Monkey M: R=0.29***

2) The analysis of neural onset times is interesting and constitutes a novel major finding of the paper. However, the results could potentially be affected by the fact that contralateral perturbation responses tended to be larger than ipsilateral responses. If a neuron's response has the same time course in two conditions, but reaches a higher firing rate in one condition, then a set threshold for detecting the response time will yield an earlier for the higher firing rate condition. Does the earlier response time for contralateral movements still hold if controlling for the magnitude of firing rate changes? A subset analysis on neurons with matched response magnitudes (or a scatter plot of onset difference vs. response magnitude ratio) may be useful to address this concern.

We agree that the magnitude differences may result in earlier onsets for the contralateral activity. We addressed this by first investigating if there was a relationship between magnitude and onsets within a context (i.e. contra or ipsi). For both monkeys we found no significant correlations between magnitude and onset timing for the contralateral (Monkey P: r=-0.08, Monkey M: r=0.13) and ipsilateral activity (Monkey P: r=0.12, Monkey M: r=0.004). Next we compared the difference in onsets with the magnitude ratios between contralateral and ipsilateral activity as shown in Figure 7D, H. The data clearly show little relationship between magnitude and onset as the Pearson’s correlation coefficients was -0.08 and -0.07 for Monkeys P and M, respectively. Lastly, we included Figure 7—figure supplement 1, which examines a subset of neurons with an absolute difference in magnitude that was less than <20Hz/Nm. This generated a subset of neurons with an equal number of neurons that had larger contralateral than ipsilateral responses and vice versa (Figure 7—figure supplement 1A, D). Comparing the onsets between contralateral and ipsilateral activity still clearly showed a bias for earlier contralateral activity than ipsilateral activity (Figure 7—figure supplement 1B, C). For Monkey M, we found the distribution of onset differences was still significantly shifted to the left indicating faster contralateral responses(Wilcoxon signed-rank test: z=2.8 p<0.01). For Monkey P we saw a similar trend for faster contralateral responses however the shift was not significant (z=1.4 p=0.17). These data suggest the earlier onset of the contralateral activity was not simply due to the larger change in activity. These figures and analyses have been added to the Results.

3) The control analysis presented in Figure 8 should be clarified, revised or removed. One major concern was that the result presented in Figure 8B and C (larger spread of ipsilateral weights as compared to contralateral weights) may be statistical artifact related to the fact that the variances (or sums-of-squares) of the regressors (velocity of the contralateral and ipsilateral arm) are vastly different. The predicted variance of the regression coefficients is lawfully related to inverse of the variance of the regressor. More precisely in the context of multiple regression with design matrix X, the var(b) is proportional to inv(X'*X). Additionally, the ipsilateral movement is correlated to that on the contralateral arm, further increasing the variance of the weights. Thus arguments of "higher sensitivity" or " context dependent selectivity" need to be carefully reinvestigated.

We thank the reviewers for raising this concern regarding our control analysis. Our intentions with this analysis was to demonstrate that if the M1 activity we observed during ipsilateral perturbations were the result of motion in the non-perturbed (contralateral) hand, then M1 activity would be exhibiting an increased sensitivity to limb motion. Supporting this was the fact that M1 activity for ipsilateral loads was twice as small as for contralateral loads while kinematic motion was ten times smaller. We thought regressing neural activity on to the limb motion would provide supporting evidence of this. However, when we re-examined the analysis, we found we could predict the unperturbed limb’s motion from the perturbed limb’s motion for ipsilateral loads (R^2^ range: 0.81 – 0.99). This confound explains how the perturbed and non-perturbed limb could account for approximately the same amount of M1 activity variance (as noted by reviewer 2).

As a result, we think removing this analysis from the paper would be best. Instead, we have added to the Discussion “…we demonstrated for ipsilateral loads that M1 activity was approximately half the size of the activity for contralateral loads. Correspondingly, we showed for ipsilateral loads that the kinematic motion of the contralateral (unperturbed) hand was approximately ten times smaller than for contralateral loads. Thus, M1’s sensitivity to contralateral hand motion would have to increase substantially when not engaged in the task, inconsistent with Omrani et al., 2014, 2016.”

Overall, the reviewers also commented that the clarity of the writing needs to be improved.Reviewer #1:[…] I have a few major suggestions, shared below, which I envision the authors should be able to fully address without major new experiments or analyses. If the authors address all of these concerns, I would be quite supportive of this paper's publication at eLife.1) The authors should provide more information about how recordings were combined across days. The Materials and methods state (subsection “Neural, EMG and kinematic recordings”) that neurons were recorded from 3 behavioral sessions spaced 2-3 months apart. Because the electrodes of chronically-implanted Utah arrays are not moved between sessions, there is always a concern that recordings from different days could be picking up an overlapping set of units. Were any controls performed to ensure that no neurons were duplicated in the dataset?

Please see Essential revision 1.

2) The analysis of neural onset times could potentially be affected by the fact that contralateral perturbation responses tended to be larger than ipsilateral responses. If a neuron's response has the same time course in two conditions, but reaches a higher firing rate in one condition, then a set threshold for detecting the response time will yield an earlier for the higher firing rate condition. Does the earlier response time for contralateral movements still hold if controlling for the magnitude of firing rate changes?

Please see Essential revision 2.

Reviewer #2:This manuscript describes a study of the activity in M1 in response to both ipsilateral and contralateral limb movements imposed by an exoskeleton. The title and Abstract are strongly focused on the second half of the study, the idea that the subspaces occupied by the activity driven by the two different arms are independent of one another. The types of methods they employ are being used in an increasing number of studies to investigate a wide range of properties of the networks of neurons that can be recorded by chronically implanted electrode arrays. The first (and lengthier) part of the study uses more traditional methods to analyze the basic properties of single neurons in response to perturbations of the two limbs. I edited the first part rather closely, until I got to the control section. I found the description to be unnecessarily hard to follow, but much more importantly, assuming I understood what they did, it raised major concerns in my mind, rather than alleviating any. Unless these can be adequately explained away, this strikes me as a tremendous amount of description and analysis of signals, the origin of which, is pretty unclear.Comments related to my major concern:“contralateral limb motion that was transferred through the body”. Meaning motion of the trunk with the arm stable, that causes shoulder rotation? Seems awfully unlikely. Doesn't the Kinarm constrain trunk motion?

Perhaps we were being too careful and picky. The exoskeleton has two arm troughs that support the upper and lower forearm and joints aligned with the shoulder and elbow. The only trunk support was a back support and the monkey was sitting on a perch. In theory, motion should not be transferred to the other limb, however there is always a chance that there is some motion between the arm and the troughs that leads to motion of the trunk and other limb.

Supporting this is the observation that following a perturbation, changes in hand motion in the non-perturbed limb starts <100ms after the perturbation (Figure 4—figure supplement 1). In theory, the motion may have been generated by small muscular contractions in the non-perturbed limb, rather than mechanical transfer through the body. However, we failed to observe any changes in muscle activity in the non-perturbed limb. We think the more reasonable explanation is trunk motion caused by the perturbed limb. We have added this argument to the Discussion (third paragraph). We also include Figure 4—figure supplement 1 showing the motion of the non-perturbed limb on smaller temporal and spatial scales.

“We addressed this by fitting the average neural activity when contralateral loads were applied to the average contralateral hand velocity in the perturbation epoch (data not shown).” I must not be following this, as I don't follow how this addresses the concern. You're trying to predict M1 from ipsilateral hand motion using a mapping derived from contra hand motion? I can't imagine that working well, or how it serves as a control. I gather the Kinarm was not being used to stabilize the non-moving limb?

Please see Essential revision 3. Also, we have added to the Materials and methods that the non-perturbed limb was not constrained or stabilized by the exoskeleton (subsection “Animal and apparatus”, second paragraph).

“Contralateral and ipsilateral activity evoked a change in firing rate over a similar range for this neuron, but contralateral loads evoked larger hand speeds.” I find this to be really confusing. Activity doesn't "evoke" firing rate; that's how it's measured. Is that what you mean? Furthermore, is the (essentially zero) ipsilateral hand movement that which resulted from (was "evoked by") loads applied to the opposite limb? Unless I'm missing something pretty basic, this section is much more complicated than necessary. The opposite hand simply doesn't move. What more is there than that?

Please see Essential revision 3.

“We fit the evoked neural activity when ipsilateral loads were applied to the corresponding contralateral hand motion.” I confess I'm completely lost. What is meant by, "ipsilateral loads were applied to the corresponding contralateral hand motion." Does it mean mapping from the (tiny) non-moving hand speed to contralateral M1 activity?

Please see Essential revision 3.

“indicating the size of the ipsilateral weights were larger than the corresponding contralateral weights.” In fact, they differ by more than an order of magnitude, presumably because the non-perturbed ("contralateral") hand was moving so slowly. And yet the (tiny) hand movement accompanying loads on the opposite side of the body predicted contralateral M1 well? This causes major concerns about the validity of the entire study, if M1 activity is better predicted by the tiny contralateral hand movements than by the ipsilateral movements (albeit, that generalization was doomed from the outset). I wonder how well you can decode ipsilateral and contralateral limb movement from M1 activity when perturbations drive the ipsilateral arm? If they are of similar quality, or contra decoding is better, it would be a huge concern.

Please see Essential revision 3. Also, we tried a preliminary decoder that predicted the limb motion of the perturbed and non-perturbed limb for the first 300ms post-perturbation using the recorded neural activity (similar to Stavisky et al., 2017). When we perturbed the ipsilateral limb, we decoded the motion of the ipsilateral limb better (Monkey P/M, R^2^ =40/35%, leave-one-out cross-validation) than the unperturbed contralateral limb (R^2^ = 20/-2%) highlighting that the activity is likely related to the perturbed ipsilateral limb. However, the decoder performance was fairly poor (40/35%), particularly given that we used trial-averaged activity. We think this reflects that the optimal lead/lag between decoder and limb motion is complex during a perturbation. The earliest feedback response after the perturbation is driven by feedback from the limb (neural activity lags limb motion), whereas activity ~50ms after the perturbation begins to also reflect motor responses to counter the load (neural activity will begin to lead and drive limb motion, Pruszynski et al., 2011, 2014). Thus, in a feedback corrective response there is a complex interaction between sensory motion and neural processing that is not captured well with a simple decoder model. Thus, we think that this analysis is beyond the scope of the current study and did not include the decoder results in the article.